# KerZOO: Kernel Function Informed Zeroth-Order Optimization for Accurate and Accelerated LLM Fine-Tuning

## Abstract

Large language models (LLMs) have demonstrated impressive capabilities across numerous NLP tasks. Nevertheless, conventional first-order fine-tuning techniques impose heavy memory demands, creating practical obstacles to real-world applications. Zeroth-order (ZO) optimization has recently emerged as a promising memory-efficient alternative, as it circumvents the need for backpropagation by estimating gradients solely through forward passes—making it particularly suitable for resource-limited environments. Despite its efficiency, ZO optimization suffers from gradient estimation bias, which significantly hinders convergence speed. To address this, we analytically identify and characterize the lower-order bias introduced during ZO-based gradient estimation in LLM fine-tuning. Motivated by tools in mathematical physics, we introduce a kernel-function-based ZO framework aimed at mitigating this bias and improving optimization stability. KerZOO achieves comparable or superior performance to existing ZO baselines in both full-parameter and parameter-efficient fine-tuning settings of LLMs, while significantly reducing the number of iterations required to reach convergence. For example, KerZOO reduces total GPU training hours by as much as 74% and 44% on WSC and MultiRC datasets in fine-tuning OPT-2.7B model and can exceed the MeZO baseline by 2.9% and 2.6% in accuracy. We show that the kernel function is an effective avenue for reducing estimation bias in ZO methods.

## 1 Introduction

The fine-tuning of pre-trained large language models (LLMs) for downstream tasks has attracted growing interest (Hu et al., 2021; Dettmers et al., 2023; Gu et al., 2021). As model sizes continue to grow, full parameter fine-tuning (FT) becomes increasingly memory intensive, presenting significant computational challenges (Tan et al., 2025; Zhao et al., 2024b). To address GPU memory constraints associated with full fine-tuning, researchers have proposed parameter-efficient fine-tuning (PEFT) methods (Hu et al., 2022; Li & Liang, 2021; Dettmers et al., 2023; Zhao et al., 2024a). These approaches update only a small subset of (additional) parameters, substantially reducing computational and storage costs while maintaining performance comparable to that of fully fine-tuned models.

Adaptive first-order optimizers such as Adam and AdamW (Loshchilov & Hutter, 2017; Kingma & Ba, 2014) are widely used for fine-tuning large language models (LLMs). However, these optimizers still incur substantial memory consumption, primarily due to the backpropagation process required for gradient computation in first-order optimization. To address the limitations, zeroth-order (ZO) optimization has emerged as a promising memory-efficient paradigm for LLM fine-tuning, attracting significant attention (Zhang et al., 2024; Zhao et al., 2024b; Malladi et al., 2023). Through fine-tuning large language models with only forward pass, it can achieve substantial memory reductions, makes it feasible to train and store LLMs on consumer hardware without the need for backpropagation.

However, while zeroth-order optimization methods significantly reduce memory consumption, they achieve this at the cost of slower convergence and reduced accuracy. Compared to first-order methods, ZO methods exhibit markedly inferior performance in terms of both convergence speed and time (GPU hours) (Tan et al., 2025). Previous studies have analyzed the underlying causes of this issue, attributing it primarily to two key factors: (1) The parameters of LLMs often exhibit heterogeneous

curvature across different dimensions (Sagun et al., 2016; Ghorbani et al., 2019; Zhang et al., 2020). Such significant variation in second-order derivatives can cause ZO methods to converge toward saddle points, substantially impeding overall optimization convergence. (2) Zeroth-order optimization methods estimate gradients via randomly sampled perturbations (Malladi et al., 2023; Gautam et al., 2024). Consequently, when the sampled directions are suboptimal, the induced lower-order bias in the gradient estimation can significantly hinder convergence speed and degrade overall accuracy (Lobanov & Gasnikov, 2023; Akhavan et al., 2024).

The first issue mentioned above has been analyzed and partially addressed in HIZOO (Zhao et al., 2024b). In this work, we focus on the second issue of the lower-order bias introduced by random perturbations in zeroth-order optimization (ZO). To mitigate the issue, we propose KerZOO, a kernel function informed Zeroth-Order Optimization approach, to mitigate the lower-order estimation bias, thus improving the convergence speed during LLMs fine-tuning. Experimental results show that KerZOO can dramatically reduce the required training steps, making the optimization process more efficient without sacrificing accuracy compared with the baselines. We summarize our contributions as follows:

- We provide theoretical analysis on how existing zeroth-order method can introduce lower-order bias in gradient estimation of LLMs fine-tuning.

- For the first time, we show that the kernel function can help mitigate the lower-order bias issue in zeroth-order optimization for LLMs fine-tuning with theoretical analysis. Moreover, we provide the design principle of the kernel function, aiming to remove the lower-order bias in LLMs fine-tuning, thus improving the convergence speed.

- We conduct extensive experiments across different models including encoder-only model (e.g., RoBERTa-large) and autoregressive language models (e.g., OPT and LLaMA). Experimental results show that KerZOO can achieve high accuracy and faster convergence. For example, for OPT-2.7B model fine-tuning, we can achieve up to 2.9% and 2.6% higher accuracy with 74% and 44% less GPU hours in convergence compared with the baseline on WSC and MultiRC datasets, respectively.

## 2 RELATED WORK

### 2.1 MEMORY-EFFICIENT LLM FINE-TUNING

Fine-tuning pre-trained models (Devlin, 2018; Liu et al., 2019; Chen et al., 2022; Radford et al., 2021; Singh et al., 2022) has emerged as an effective strategy for leveraging previously acquired representations. However, it remains resource-intensive especially with the scaling model size, prompting the development of more memory-efficient alternatives. Parameter-efficient fine-tuning (PEFT) techniques, including LoRA (Hu et al., 2021) and prefix tuning (Li & Liang, 2021), address this challenge by modifying only a limited portion of model parameters , thereby retaining most of the original pre-trained weights and the knowledge they encode. Low-rank adaptation methods which exemplified by LoRA, have shown strong performance by introducing low-rank updates that are lightweight yet effective. Building on this, DoRA (Liu et al., 2024a) further decomposes pre-trained weights into direction and magnitude components to improve both expressiveness and training stability. GaLore (Zhao et al., 2024a) introduces gradient low-rank projection to enable full-parameter training while maintaining the memory efficiency characteristic of low-rank updates. In addition to low-rank techniques, quantization has become a key strategy for reducing resource usage. SmoothQuant (Xiao et al., 2023) further optimizes mixed-precision training by smoothing out activation outliers through offline scaling, shifting quantization difficulty from activations to weights. Outlier Suppression+ (Wei et al., 2023) enhances low-bit (e.g., int4) quantization by applying channel-wise shifting and scaling to reduce asymmetry and variance in activation distributions.

### 2.2 ZEROTH-ORDER OPTIMIZATION AND ACCELERATION

Zeroth-order optimization, a classical optimization schema that uses only differences of loss values for gradient estimation, has attracted significant attention in the machine learning community (Chen et al., 2017; Ye et al., 2018; Verma et al., 2023; Chen et al., 2023; Malladi et al., 2023). Unlike conventional first-order optimization, which computes the gradient via cumbersome backpropagation, the ZO method can, in principle, update the model with just forward passes.

Recently, the ZO method has been proven to be efficient in solving the significant memory limitation in large-scale LLMs fine-tuning (Malladi et al., 2023; Liu et al., 2024c; Tang et al., 2024; Zhao et al., 2024b). However, ZO optimization often converges more slowly than FO approaches, primarily due to the noise from its randomized estimators. A variety of strategies have been proposed to improve the efficiency of ZO optimization. (Liu et al., 2018) introduces ZO-SVRG by integrating variance reduction methods (Johnson & Zhang, 2013), helping to stabilize gradient estimates. (Shu et al., 2023) employs Gaussian process surrogates to approximate the objective landscape, thereby reducing query overhead and enabling denser sampling. (Li et al., 2024) advances the idea of pretrained optimizers, which transfer knowledge across tasks to enable rapid zero-shot adaptation within a few ZO fine-tuning steps. In parallel, (Zhao et al., 2024b) proposes HiZOO, a framework that incorporates second-order information via estimated Hessians to guide more effective updates. Despite these efforts, adapting ZO methods to LLMs still faces challenges. Many accelerator were designed for small-scale tasks and lack scalability. Moreover, as recent work suggests single-step tuning can suffice (Malladi et al., 2023), the focus needs to shifts from query efficiency to stability. Additionally, some accelerators also compromise ZO's key benefits in low memory usage and high throughput. These challenges underscore the need for scalable, efficient ZO strategies for LLM fine-tuning.

## 3 PROPOSED METHOD

**Motivation.** In existing works in zeroth-order optimization for LLMs fine-tuning, perturbations are directly applied to the to be optimized variables, without considering the bias introduced by the perturbations However, due to the inherent stochastic nature of zeroth-order methods, excessive bias can significantly slow down convergence and degrade optimization performance (Akhavan et al., 2024; Bach & Perchet, 2016; Ghadimi & Lan, 2013). Taking the inspiration, we propose a zeroth-order optimization method incorporating kernel functions to mitigate the lower-order estimation bias, thus improving convergence speed during LLMs fine-tuning.

### 3.1 REVISITING ZEROTH-ORDER OPTIMIZATION FOR LLMS

Given a large language model with parameters $\boldsymbol{\theta} \in \mathbb{R}^d$ and loss function $\mathcal{L}$, we can use ZO method to estimate the gradient on a minibatch $\mathcal{B}_t$ at the iteration step $t$ , based on the concepts of sampling and differencing, as shown below:

$$\nabla\mathcal{L}(\boldsymbol{\theta}_t; \mathcal{B}_t) = \frac{\mathcal{L}(\boldsymbol{\theta}_t + \epsilon\boldsymbol{u}; \mathcal{B}_t) - \mathcal{L}(\boldsymbol{\theta}_t - \epsilon\boldsymbol{u}; \mathcal{B}_t)}{2\epsilon}\boldsymbol{u} \tag{1}$$

where $\boldsymbol{u} \in \mathbb{R}^d$ and $\boldsymbol{u}$ is a unit vector sampled from unit Gaussian sphere (i.e., $u = \frac{v}{\|v\|}$, where $v \sim \mathcal{N}(0, I_d)$), $\epsilon$ is the perturbation scale. We can also use multiple sampled $\boldsymbol{u}$ to get the $n$-average gradient:

$$\nabla\mathcal{L}(\boldsymbol{\theta}_t; \mathcal{B}_t) = \frac{1}{n}\sum_{i=1}^{n}[\frac{\mathcal{L}(\boldsymbol{\theta}_t + \epsilon\boldsymbol{u_i}; \mathcal{B}_t) - \mathcal{L}(\boldsymbol{\theta}_t - \epsilon\boldsymbol{u_i}; \mathcal{B}_t)}{2\epsilon}\boldsymbol{u_i}] \tag{2}$$

Given the learning rate $\eta$ and the mini-batch data $\mathcal{B}_t$ at $t$-th iteration, once the estimated gradient $\nabla\mathcal{L}(\boldsymbol{\theta}; \mathcal{B}_t)$ is obtained, then ZO-SGD updates the parameters as follows:

$$\boldsymbol{\theta}_{t+1} = \boldsymbol{\theta}_t - \eta\nabla\mathcal{L}(\boldsymbol{\theta}_t; \mathcal{B}_t) \tag{3}$$

### 3.2 PROBLEM IN EXISTING ZEROTH-ORDER OPTIMIZATION FOR LLMS

Since ZO methods estimate gradients based on randomly sampled perturbations $\boldsymbol{u}$, suboptimal directions can introduce substantial lower-order bias in the gradient estimates, thereby reducing estimation accuracy and hindering convergence speed (Lobanov & Gasnikov, 2023; Akhavan et al., 2024). To further analyze the issue, we apply Taylor expansion on different terms in equation 1. On the same minibatch, we have:

$$\mathcal{L}(\boldsymbol{\theta} + \epsilon\boldsymbol{u}) = \mathcal{L}(\boldsymbol{\theta}) + \epsilon\langle\nabla\mathcal{L}(\boldsymbol{\theta}), \boldsymbol{u}\rangle + \frac{(\epsilon)^2}{2}\boldsymbol{u}^\top\nabla^2 f(x)\boldsymbol{u} + \frac{(\epsilon)^3}{6}D^3\mathcal{L}(\boldsymbol{\theta})[\boldsymbol{u}, \boldsymbol{u}, \boldsymbol{u}] + O(\epsilon^4) \tag{4}$$

$$\mathcal{L}(\boldsymbol{\theta} - \epsilon\boldsymbol{u}) = \mathcal{L}(\boldsymbol{\theta}) - \epsilon\langle\nabla\mathcal{L}(\boldsymbol{\theta}), \boldsymbol{u}\rangle + \frac{(\epsilon)^2}{2}\boldsymbol{u}^\top\nabla^2 f(x)\boldsymbol{u} - \frac{(\epsilon)^3}{6}D^3\mathcal{L}(\boldsymbol{\theta})[\boldsymbol{u}, \boldsymbol{u}, \boldsymbol{u}] + O(\epsilon^4) \quad (5)$$

where $\nabla\mathcal{L}(\boldsymbol{\theta})$ is the gradient at $\boldsymbol{\theta}$, $\nabla^2\mathcal{L}(\boldsymbol{\theta})$ is the Hessian matrix at $\boldsymbol{\theta}$, and $D^3\mathcal{L}(\boldsymbol{\theta})[\boldsymbol{u}, \boldsymbol{u}, \boldsymbol{u}]$ denotes the third-order directional derivative of $\mathcal{L}$ along $\boldsymbol{u}$ three times. Taking the difference, we have:

$$\mathcal{L}(\boldsymbol{\theta} + \epsilon\boldsymbol{u}) - \mathcal{L}(\boldsymbol{\theta} - \epsilon\boldsymbol{u}) = 2\epsilon\langle\nabla\mathcal{L}(\boldsymbol{\theta}), \boldsymbol{u}\rangle + \frac{(\epsilon)^3}{3}D^3\nabla\mathcal{L}(\boldsymbol{\theta})[\boldsymbol{u}, \boldsymbol{u}, \boldsymbol{u}] + O(\epsilon^4) \quad (6)$$

Therefore, the equation 1 can be expressed as:

$$\frac{\mathcal{L}(\boldsymbol{\theta} + \epsilon\boldsymbol{u}) - \mathcal{L}(\boldsymbol{\theta} - \epsilon\boldsymbol{u})}{2\epsilon} \cdot \boldsymbol{u} = \langle\nabla\mathcal{L}(\boldsymbol{\theta}), \boldsymbol{u}\rangle\boldsymbol{u} + \frac{(\epsilon)^2}{6}D^3\nabla\mathcal{L}(\boldsymbol{\theta})[\boldsymbol{u}, \boldsymbol{u}, \boldsymbol{u}] \cdot \boldsymbol{u} + O(\epsilon^4) \quad (7)$$

Taking the expectation (noting that $\mathbb{E}[\boldsymbol{u}\boldsymbol{u}^\top] = \frac{1}{d}I_d$), we have:

$$\mathbb{E}[\frac{\mathcal{L}(\boldsymbol{\theta} + \epsilon\boldsymbol{u}) - \mathcal{L}(\boldsymbol{\theta} - \epsilon\boldsymbol{u})}{2\epsilon} \cdot \boldsymbol{u}] = \frac{1}{d}\nabla\mathcal{L}(\boldsymbol{\theta}) + \mathbb{E}[\frac{(\epsilon)^2}{6}D^3\nabla\mathcal{L}(\boldsymbol{\theta})[\boldsymbol{u}, \boldsymbol{u}, \boldsymbol{u}] \cdot \boldsymbol{u}] + O(\epsilon^4) \quad (8)$$

As shown in the above formulation, zeroth-order methods yield estimated gradients containing higher-order bias terms $\mathbb{E}[\frac{(\epsilon)^2}{6}D^3\nabla\mathcal{L}(\boldsymbol{\theta})[\boldsymbol{u}, \boldsymbol{u}, \boldsymbol{u}] \cdot \boldsymbol{u}] + O(\epsilon^4)$, which are highly sensitive to the choice of perturbation direction $\boldsymbol{u}$, and a suboptimal sampling of $\boldsymbol{u}$ can result in large bias, leading to subpotimal gradient estimation and slow convergence.

### 3.3 KERNEL FUNCTION INFORMED ZEROTH-ORDER OPTIMIZATION

Motivated by kernel smoothing techniques in mathematical physics, which are widely used to reduce estimation bias (Nesterov & Spokoiny, 2017; Bach & Perchet, 2016; Akhavan et al., 2024), we propose a kernel function informed zeroth-order optimization method for LLMs fine-tuning to address the challenge in gradient estimation arising from the high dimensionality of LLMs in which the randomly sampled perturbations exhibit different values in different directions.

To control the perturbation magnitude, we incorporate a random scalar variable $r$ in equation 1 and use $\hat{g}$ to denote the estimated gradient as:

$$\hat{g} = \frac{\mathcal{L}(\boldsymbol{\theta} + \epsilon r\boldsymbol{u}) - \mathcal{L}(\boldsymbol{\theta} - \epsilon r\boldsymbol{u})}{2\epsilon}\boldsymbol{u} \quad (9)$$

where $\boldsymbol{u}$ is a random unit vector (e.g., uniformly sampled from the unit Gaussian sphere), and $\epsilon > 0$ is a small step size denoted as perturbation scale.

Assuming loss function $\mathcal{L}$ is at least third-order differentiable, by performing a Taylor expansion on $\mathcal{L}(\boldsymbol{\theta} \pm \epsilon r\boldsymbol{u})$, we have:

$$\mathcal{L}(\boldsymbol{\theta} + \epsilon r\boldsymbol{u}) = \mathcal{L}(\boldsymbol{\theta}) + \epsilon r\langle\nabla\mathcal{L}(\boldsymbol{\theta}), \boldsymbol{u}\rangle + \frac{(\epsilon r)^2}{2}\boldsymbol{u}^\top\nabla^2 f(x)\boldsymbol{u} + \frac{(\epsilon r)^3}{6}D^3\mathcal{L}(\boldsymbol{\theta})[\boldsymbol{u}, \boldsymbol{u}, \boldsymbol{u}] + O(\epsilon^4) \quad (10)$$

$$\mathcal{L}(\boldsymbol{\theta} - \epsilon r\boldsymbol{u}) = \mathcal{L}(\boldsymbol{\theta}) - \epsilon r\langle\nabla\mathcal{L}(\boldsymbol{\theta}), \boldsymbol{u}\rangle + \frac{(\epsilon r)^2}{2}\boldsymbol{u}^\top\nabla^2 f(x)\boldsymbol{u} - \frac{(\epsilon r)^3}{6}D^3\mathcal{L}(\boldsymbol{\theta})[\boldsymbol{u}, \boldsymbol{u}, \boldsymbol{u}] + O(\epsilon^4) \quad (11)$$

Taking the difference between the two expressions, we get:

$$\mathcal{L}(\boldsymbol{\theta} + \epsilon r\boldsymbol{u}) - \mathcal{L}(\boldsymbol{\theta} - \epsilon r\boldsymbol{u}) = 2\epsilon r\langle\nabla\mathcal{L}(\boldsymbol{\theta}), \boldsymbol{u}\rangle + \frac{(\epsilon r)^3}{3}D^3\nabla\mathcal{L}(\boldsymbol{\theta})[\boldsymbol{u}, \boldsymbol{u}, \boldsymbol{u}] + O(\epsilon^4) \quad (12)$$

Combining Equation 12 and Equation 9, we have:

$$\hat{g} = \frac{\mathcal{L}(\boldsymbol{\theta} + \epsilon r\boldsymbol{u}) - \mathcal{L}(\boldsymbol{\theta} - \epsilon r\boldsymbol{u})}{2\epsilon} \cdot \boldsymbol{u} = r\langle\nabla\mathcal{L}(\boldsymbol{\theta}), \boldsymbol{u}\rangle\boldsymbol{u} + \frac{(\epsilon)^2 r^3}{6}D^3\nabla\mathcal{L}(\boldsymbol{\theta})[\boldsymbol{u}, \boldsymbol{u}, \boldsymbol{u}] \cdot \boldsymbol{u} + O(\epsilon^4) \quad (13)$$

where the first term $r\langle\nabla f(x), u\rangle u$ can be used to approximate the true gradient $\mathcal{L}(\boldsymbol{\theta})$, and the second term $\frac{(\epsilon)^2 r^3}{6}D^3\nabla\mathcal{L}(\boldsymbol{\theta})[\boldsymbol{u}, \boldsymbol{u}, \boldsymbol{u}] \cdot \boldsymbol{u}$ introduces a leading bias of order $O(\epsilon^2)$. To reduce the second-order bias, we introduce a kernel function $K(r)$, and modify the gradient estimator as:

$$\hat{g}_K = \frac{\mathcal{L}(\boldsymbol{\theta} + \epsilon r\boldsymbol{u}) - \mathcal{L}(\boldsymbol{\theta} - \epsilon r\boldsymbol{u})}{2\epsilon}K(r)\boldsymbol{u} \quad (14)$$

---

**Algorithm 1** KerZOO

---

1: **Inputs:** Starting points of the LLM parameters $\boldsymbol{\theta}_0^{ag} = \boldsymbol{\theta}_0 \in \mathbb{R}^d$ ($\boldsymbol{\theta}_0^{ag}$ is an intermediate variable as same as the model parameters $\boldsymbol{\theta}_0$, and it is also updated at each iteration), number of iterations $N$, perturbation number $n$, perturbation scale $\epsilon > 0$, kernel $K(\cdot)$, learning rate $\eta$, iteration constant $\beta_0 = 1$, gradient clip constant $R$.

2: **for** $t = 0$ to $N - 1$ **do**

3: $\quad \beta_t = 1 + \frac{t}{6}$

4: $\quad \boldsymbol{\theta}_t^{md} = \beta_t^{-1}\boldsymbol{\theta}_t + (1 - \beta_t^{-1})\boldsymbol{\theta}_t^{ag}$

5: $\quad$ Sample random perturbation $\boldsymbol{u}_i \sim \mathcal{N}(0, I_d)$, $r_i \sim \text{Uniform}[-1, 1]$ for $i = 1, \ldots, n$

6: $\quad$ Compute batched gradient approximation:

$$\boldsymbol{g}_K^t = \mathbb{E}[\hat{g}_K^t] = \frac{1}{n}\sum_{i=1}^n \frac{\mathcal{L}(\boldsymbol{\theta}_t + \epsilon r_i \boldsymbol{u}_i) - \mathcal{L}(\boldsymbol{\theta}_t - \epsilon r_i \boldsymbol{u}_i)}{2\epsilon}K(r_i)\boldsymbol{u}_i$$

7: $\quad \bar{\boldsymbol{\theta}}_{t+1} = \boldsymbol{\theta}_t - \eta\boldsymbol{g}_K^t$

8: $\quad \boldsymbol{\theta}_{t+1} = \min\left\{1, \frac{R}{\|\bar{\boldsymbol{\theta}}_{t+1}\|}\right\}\bar{\boldsymbol{\theta}}_{t+1}$

9: $\quad \boldsymbol{\theta}_{t+1}^{ag} = \beta_t^{-1}\boldsymbol{\theta}_{t+1} + (1 - \beta_t^{-1})\boldsymbol{\theta}_t^{ag}$

10: **end for**

11: **return** $\boldsymbol{\theta}_N^{ag}$

---

Applying the Taylor expansions, we have:

$$\hat{g}_K = (r\langle\nabla\mathcal{L}(\boldsymbol{\theta}), \boldsymbol{u}\rangle K(r))\boldsymbol{u} + \frac{(\epsilon)^2 r^3}{6}D^3\nabla\mathcal{L}(\boldsymbol{\theta})[\boldsymbol{u}, \boldsymbol{u}, \boldsymbol{u}]K(r)\boldsymbol{u} + O(\epsilon^4) \tag{15}$$

Taking the expectation of both sides of Equation 15, we have:

$$\mathbb{E}[\hat{g}_K] = \mathbb{E}[rK(r)]\frac{1}{d}\nabla\mathcal{L}(\boldsymbol{\theta}) + \mathbb{E}[r^3 K(r)]\mathbb{E}[\frac{(\epsilon)^2}{6}D^3\nabla\mathcal{L}(\boldsymbol{\theta})[\boldsymbol{u}, \boldsymbol{u}, \boldsymbol{u}]\boldsymbol{u}] + O(\epsilon^4) \tag{16}$$

To remove the lower-order bias (i.e., the second term in Equation 16), we develop the following kernel function $K(r)$ design principle:

- **First-moment condition:** $\mathbb{E}[rK(r)] = C$ ($C$ is a constant), ensuring that the estimator remains approximately unbiased for the true gradient;

- **Third-moment condition:** $\mathbb{E}[r^3 K(r)] = 0$, eliminating the leading second-order bias term.

Based on the kernel function design principle, Equation 16 becomes $\mathbb{E}[\hat{g}_K] = \frac{C}{d}\nabla\mathcal{L}(\boldsymbol{\theta}) + O(\epsilon^4)$.

By carefully constructing $K(r)$ to satisfy these moment conditions (see Section 3.4 for details), we can effectively remove the second-order (lower-order) bias, leading to a more accurate gradient estimator which have only fourth-order (higher order) bias. In our practical application, we can limit the $r$ in a smaller range as the iteration step increases (see Appendix B.1 for details). Additionally, to satisfy the condition $\mathbb{E}[r^3 K(r)] = 0$ and $\mathbb{E}[rK(r)] = C$, it is essential to perform multiple perturbations so that the expectation becomes statistically meaningful. Our kernel function informed zeroth-order optimization can be found in Algorithm 1.

### 3.4 KERNEL FUNCTION DESIGN

Here we describe the details of the kernel function design. According to (Polyak & Tsybakov, 1990), we consider $r$ uniformly distributed in $[-1, 1]$, then we may choose $K_\beta(r) = C \cdot \sum_{m=0}^{\beta} p_m'(0)p_m(r)$.

$$K(r) = C \cdot \sum_{m=0}^{\beta} p_m'(0)p_m(r) \tag{17}$$

In the expression,

$$p_m(u) = \sqrt{2m+1}L_m(u) \tag{18}$$

$L_m(u)$ denotes the $m$-th Legendre polynomial. $p'(0)$ denotes the derivative of the polynomial at the point $m = 0$. $\beta$ represents the order of the polynomial, corresponding to the highest power $s$ that the expression $\mathbb{E}[r^s K(r)] = 0$ can accommodate. $C$ represents a constant.

For example, we have the following values for $\beta \in \{1, 3, 5\}$:

$$K_1(r) = C \cdot 3r$$
$$K_3(r) = C \cdot \frac{15}{4} r(5 - 7r^2)$$
$$K_5(r) = C \cdot \frac{195}{64} r(99r^4 - 126r^2 + 35)$$

Taking $\beta = 3$ as an example:

$$\mathbb{E}[rK_3(r)] = \int_{-1}^{1} \frac{15C}{4} r^2(5 - 7r^2) \cdot \rho(r) \, dr = C \tag{19}$$

$$\mathbb{E}[r^3 K_3(r)] = \int_{-1}^{1} \frac{195C}{64} r^4(5 - 7r^2) \cdot \rho(r) \, dr = 0 \tag{20}$$

$\rho(r)$ is the probability density function of the variable $r$ and $\rho(r) = \frac{1}{2}$ if $r \sim \mathcal{U}[-1, 1]$. As to higher order kernel function such as $K_5(r)$, we can also have $\mathbb{E}[r^5 K_5(r)] = \int_{-1}^{1} \frac{195C}{64} r^6(99r^4 - 126r^2 + 35) \, dr = 0$, removing more higer-order bias term. In our experiment, we choose to use $K_3(r)$ as our experimental kernel.

## 4 EXPERIMENTS

### 4.1 EXPERIMENTAL SETTINGS

**Models.** We experiment with both masked language models and autoregressive models. For masked language modeling, we use RoBERTa-large (Liu et al., 2019). For autoregressive models, we consider the OPT (Zhang et al., 2022) and LLaMA (Touvron et al., 2023) models. Model scales range from 355M to 6.7B parameters, including OPT-2.7B, OPT-6.7B, and LLaMA-3-3B and LLaMa-3-8B, covering medium to large-scale models.

**Tasks and Datasets.** To evaluate generalization across task formats, we include both classification and generation tasks. For RoBERTa-large, we follow few-shot classification with $k = 16$ and many-shot classification with $k = 512$ samples per class. We evaluate on 1,000 test examples. For generative models, we use datasets with a consistent 1,000/500/1,000 split for train/validation/test.

**Baselines.** We compare KerZOO against the state-of-the-art ZO optimization baselines: **MeZO** (Malladi et al., 2023): A memory-efficient ZO method based on symmetric perturbation gradient estimation. **HiZOO** (Zhao et al., 2024b): A recent ZO method incorporating approximate second-order curvature via diagonal Hessian estimation.

**Implementation Details.** All experiments are conducted on NVIDIA A100 or A6000 GPUs. For KerZOO, we set the number of perturbation directions to $n = 3$. Hyperparameters such as learning rate and batch size are in line with MeZO baseline. All reported results reflect the best configuration on the validation set.

### 4.2 RESULTS ON MEDIUM-SIZED MODEL

We assess the performance of KerZOO on RoBERTa-large across multiple classification benchmarks, including SST-2, MNLI, and RTE. We compare against existing zeroth-order (ZO) methods and explore both full-model and parameter-efficient (LoRA-based) tuning.

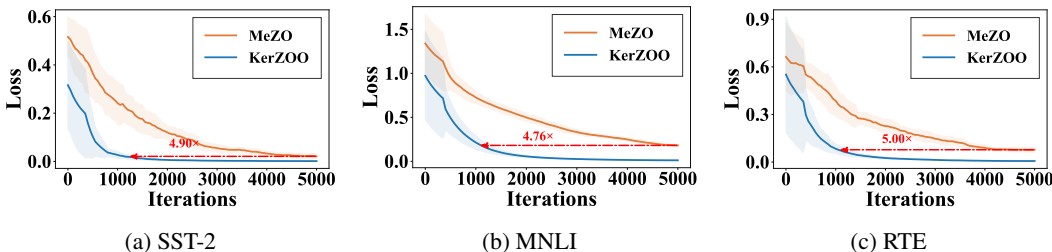

Figure 1: Training loss comparison of MeZO and KerZOO (Ours) on RoBERTa-large

Table 1: Performance of various gradient-based and gradient-free optimization methods across multiple datasets and $k$ settings on RoBERTa-large. Bold highlights best performance

| Task Type Dataset | SST-2 | SST-5 | SNLI | MNLI | RTE | TREC |
|---|---|---|---|---|---|---|
| Zero-shot | 79.0 | 35.5 | 50.2 | 48.8 | 51.4 | 32.0 |
| **Gradient-free methods:** $k = 16$ | | | | | | |
| MeZO | 90.5 (1.2) | 45.5 (2.0) | 66.0 (2.7) | 56.5 (2.5) | 59.4 (5.3) | 76.9 (2.7) |
| MeZO LoRA | 87.5 (0.7) | 41.6 (0.8) | 64.9 (0.8) | 59.5 (1.5) | 61.7 (3.2) | 58.2 (5.6) |
| KerZOO | **92.1** (0.8) | **48.5** (1.0) | **71.0** (2.2) | **63.8** (1.8) | **66.8** (3.3) | **78.2** (2.4) |
| KerZOO LoRA | 88.4 (1.0) | 42.3 (1.3) | 66.7 (1.7) | 61.1 (1.2) | 63.2 (2.8) | 59.2 (4.9) |
| **Gradient-based methods:** $k = 16$ | | | | | | |
| FT | 91.9 (1.8) | 47.5 (1.9) | 77.5 (2.6) | 70.2 (2.3) | 66.4 (7.2) | 85.0 (2.5) |
| FT LoRA | 91.4 (1.7) | 46.7 (1.1) | 74.9 (4.3) | 67.7 (1.4) | 66.1 (3.5) | 86.1 (3.3) |
| **Gradient-free methods:** $k = 512$ | | | | | | |
| MeZO | 93.3 (0.7) | 52.4 (1.2) | 83.0 (1.0) | 78.3 (0.5) | 78.6 (2.0) | 94.3 (1.3) |
| MeZO LoRA | 91.6 (0.8) | 44.8 (0.4) | 73.3 (0.6) | 66.4 (0.4) | 73.3 (1.5) | 63.8 (2.3) |
| KerZOO | **95.3** (0.5) | **53.4** (1.0) | **85.0** (1.5) | **78.3** (0.5) | **79.1** (1.8) | **96.0** (1.9) |
| KerZOO LoRA | 91.9 (0.3) | 45.0 (0.8) | 74.7 (1.1) | 65.0 (0.7) | 74.0 (2.2) | 61.0 (3.2) |
| **Gradient-based methods:** $k = 512$ | | | | | | |
| FT | 93.9 (0.7) | 55.9 (0.9) | 88.7 (0.8) | 84.4 (0.8) | 82.7 (1.4) | 97.3 (0.2) |
| FT LoRA | 94.2 (0.2) | 55.7 (0.8) | 88.3 (0.5) | 86.9 (0.6) | 83.2 (1.3) | 97.0 (0.3) |

**Faster Optimization.** KerZOO achieves a markedly faster descent in training loss compared to MeZO, as shown in Figure 1. When using only three perturbation directions, our method reduces training iterations by over 70% on average, and lowers wall-clock convergence time by 30%–40% on SST-2, MNLI, and RTE. This acceleration stems from the improved stability of our kernel-based gradient estimation.

Table 2: Results of fine-tuning OPT-2.7B on seven classification datasets and two generation datasets

| Dataset Task Type | SST-2 | RTE | CB | BoolQ | WSC | WIC | MultiRC | SQuAD | DROP |
|---|---|---|---|---|---|---|---|---|---|
| | | | | *classification* | | | | *generation* | |
| Zero-shot | 56.3 | 54.2 | 50.0 | 47.6 | 36.5 | 52.7 | 44.4 | 29.8 | 10.0 |
| FT | 94.2 | 81.2 | 82.1 | 72.2 | 63.8 | 65.8 | 71.6 | 78.4 | 30.3 |
| LoRA | 94.6 | 80.8 | 82.7 | 77.7 | 59.8 | 64.0 | 72.8 | 77.9 | 31.1 |
| MeZO | 91.6 | 63.5 | 69.6 | 67.4 | 62.5 | 59.8 | 59.4 | 63.6 | 15.3 |
| HiZOO | 90.8 | 60.6 | 70.4 | **68.0** | 60.2 | 56.6 | 54.8 | 66.0 | **18.4** |
| KerZOO | **92.6** | **65.3** | **71.4** | 67.0 | **65.4** | **60.2** | **62.0** | **66.2** | 16.0 |
| MeZO LoRA | 91.0 | 63.2 | 69.6 | 67.2 | 64.4 | 58.2 | 59.6 | 57.4 | 13.4 |
| HiZOO LoRA | 90.6 | **65.2** | 71.4 | 67.4 | 52.6 | 58.8 | 59.0 | 61.6 | 13.9 |
| KerZOO LoRA | **92.4** | 63.9 | **73.2** | 67.4 | **65.4** | **60.4** | **62.4** | 65.2 | **14.7** |

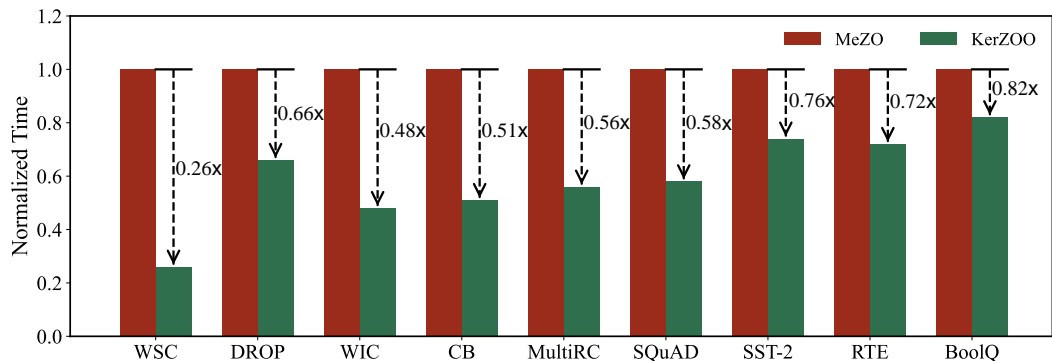

Figure 3: Comparison of GPU hours for convergence across different datasets on OPT-2.7B between MeZO and KerZOO. Results are presented as normalized time

**Improved Accuracy.** In addition to faster convergence, KerZOO can also deliver consistently higher accuracy. On the three datasets mentioned above, it improves upon MeZO by 1.7%, 7.3%, and 7.4%, respectively. The results are summarized in Table 1. In several cases such as RTE and SST-5, KerZOO even matches or exceeds the accuracy of first-order fine-tuning. We further evaluate KerZOO under the LoRA framework to test its compatibility with parameter-efficient tuning. While LoRA generally introduces a small performance drop compared to full fine-tuning, KerZOO remains competitive and continues to outperform ZO baselines in most cases. This highlights KerZOO's robustness under limited trainable parameter budgets.

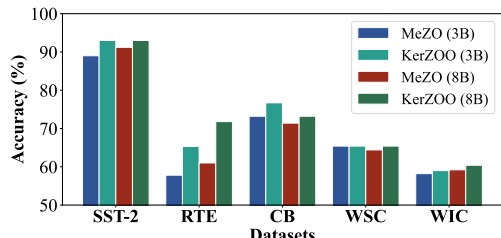

Figure 2: Results on LLaMA3-3B and LLaMA3-8B

### 4.3 RESULTS ON AUTOREGRESSIVE MODELS (OPT AND LLAMA)

We evaluate KerZOO on OPT and LLaMA series models across classification and generation tasks. The experiments span OPT-2.7B, OPT-6.7B, and LLaMA3-3B/8B models. Results are reported in Table 2, Table 3, and Figure 2, respectively. We also compare GPU time efficiency and memory usage on the SQuAD dataset in Table 4.

Across all model scales, KerZOO consistently reduces the training time required to reach competitive performance. As shown in Figure 3, it completes training with notably fewer update steps compared to MeZO, achieving up to a 74% reduction in computing time without reliance on gradient backpropagation. KerZOO offers consistent performance gains under both full-model fine-tuning and LoRA-based adaptation. On OPT-2.7B, it outperforms zeroth-order baselines in 6 of 7 classification tasks, and remains competitive on generation benchmarks. Table 3 shows that these trends persist when moving to larger models such as OPT-6.7B. When applied to LLaMA-3 variants, KerZOO also yields robust improvements, including about 10% accuracy gains on RTE (Figure 2).

Table 3: Experiment results on OPT-6.7B for four classification datasets and one text generation dataset (with 1000 training samples)

| Dataset | SST-2 | RTE | CB | WSC | SQuAD |
|---|---|---|---|---|---|
| Task Type | | *classification* | | | *generation* |
| MeZO | 92.2 | 69.6 | 73.2 | 60.6 | 68.9 |
| HiZOO | 90.8 | 66.3 | 71.8 | 62.1 | **71.9** |
| KerZOO | **94.0** | **71.9** | **78.6** | **65.4** | 70.7 |
| MeZO LoRA | 93.2 | 67.9 | **73.2** | 59.6 | 63.4 |
| HiZOO LoRA | 90.6 | 67.2 | 71.4 | 62.1 | 65.3 |
| KerZOO LoRA | **93.8** | **68.6** | 73.2 | **62.5** | **72.1** |

Moreover, KerZOO maintains memory efficiency compared to full fine-tuning. As detailed in Table 4, its memory footprint increases marginally but can achieve significant decrease of traning GPU hours. Compare to MeZO and HiZOO, our training time can decrease about 42% and 33% respectively in full fine-tuning on SQuAD datasets. In PEFT settings, KerZOO paired with LoRA achieves the best balance between memory cost and convergence speed — consuming 36% of the GPU hours required by MeZO while yielding higher accuracy.

## 5 ANALYSIS ON MEMORY AND TRAINING TIME

We evaluate KerZOO's memory cost and training efficiency under both full-parameter and LoRA-based tuning regimes. As shown in Table 4, compared with gradient-based methods such as full fine-tuning (73.5G) and LoRA (58.5G), KerZOO (16.9G) is more memory efficient as it can avoid storing gradients and activations. Compared with ZO-based baselines, KerZOO achieves a favorable trade-off between convergence speed and per-step overhead. While its per-step memory cost (16.9G) is slightly higher than MeZO (12.8G) due to the use of multiple perturbations, it substantially reduces the number of required iterations from 100% to only 16.9%, resulting in

Table 4: Memory and training time comparison on OPT-2.7B (SQuAD, avg. 300 tokens)

| Method | Mem. | Iter. | Hours |
|---|---|---|---|
| FT | 73.5G | 7.5% | 27.7% |
| LoRA | 58.5G | 6.3% | 11.5% |
| MeZO | 12.8G | 100.0% | 100.0% |
| HiZOO | 14.1G | 66.7% | 91.5% |
| KerZOO | 16.9G | 16.9% | 58.0% |
| MeZO+LoRA | 8.1G | 94.2% | 51.6% |
| HiZOO+LoRA | 9.1G | 80.0% | 65.7% |
| KerZOO+LoRA | 9.7G | 28.4% | 36.1% |

a total GPU hour reduction from 100.0% to 58.0%. On SQuAD, KerZOO lowers training time by over 40% compared to MeZO while maintaining competitive accuracy. For LoRA-based parameter-efficient settings, KerZOO maintains strong performance: KerZOO+LoRA reduces GPU hours to 36.1%, compared to 51.6% with MeZO+LoRA and 65.7% with HiZOO+LoRA, while using 9.7G memory. These results highlight that KerZOO offers efficient convergence with minimal memory overhead across a wide range of configurations.

## 6 IMPACT OF THE HYPERPARAMETERS IN KERNEL FUNCTION

**Order of the kernel function:** In our experiments with the OPT-2.7B model on SST-2 datasets (as shown in the Table 5), we observe that using a third-order kernel and a fifth-order kernel produces similar results, with their loss curves almost perfectly overlapping. Therefore, we conclude that a third-order kernel is sufficient for simplicity. The #Iterations in the table indicate the number of iterations needed for achieving the similar loss of MeZO.

Table 5: Selection of the kernel function order ($\beta$). The number of iterations needed to achieve a similar loss to MeZO is shown.

| Setting | #Iterations | Loss at iteration |
|---|---|---|
| MeZO | 4000 | 0.3392 |
| $\beta=1$ | 4000 | 0.4274 |
| $\beta=3$ | 1200 | 0.3361 |
| $\beta=5$ | 1200 | 0.3312 |

**Choice of $C$:** We study the impact of different choices of $C$ using OPT-2.7B model on SST-2 datasets, as shown in the Table 6, where $C_0 = 4$ is the value of C used for our main experiments. We observe that with $C = C_0/2$ or $C = C_0$, our method required reduced number of iterations than MeZO, while achieving similar loss of MeZO. It also shows that $C = C_0$ can deliver the best performance since it requires least training iterations (1200) than other two choices (i.e., $C = C_0$ and $C = 2C_0$) and can achieve smallest loss even compared with MeZO, which requires 4000 iterations for convergence.

Table 6: Selection of the constant C. The number of iterations needed for achieving a similar loss to MeZO is shown.

| Setting | #Iterations | Loss at iteration |
|---|---|---|
| MeZO | 4000 | 0.3392 |
| $C=C_0/2$ | 2600 | 0.3378 |
| $C=C_0$ | 1200 | 0.3361 |
| $C=2C_0$ | N/A | $\geq 0.3704$ |

## 7 CONCLUSION

In this work, we present a theoretical framework that characterizes the lower-order bias arising in gradient estimation when using zeroth-order optimization methods. Building on this analysis, we propose a Kernel Function Informed Zeroth-Order Optimization method, an improved ZO framework that leverages kernel function to eliminate the lower-order bias of zeroth-order gradient estimation. KerZOO offers notable improvements in training efficiency and consistently outperforms baseline approaches across a variety of tasks and different models. Meanwhile, KerZOO is fully compatible with parameter-efficient tuning (PEFT) techniques such as LoRA, allowing for additional acceleration without degrading accuracy. Looking ahead, we plan to extend this framework to other domains, with a particular focus on adapting it for LLM pruning or vision language model optimization.

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

## A   CLAIM OF LLM USAGE

In this work, large language models (LLMs) were used solely as a general-purpose writing assistant. Their role was limited to correcting grammar, fixing typographical errors, and polishing the language for clarity and readability.

## B   APPENDIX

### B.1   VARIANCE ANALYSIS OF THE ZEROTH-ORDER ESTIMATOR WITH KERNEL FUNCTION

When applying a kernel function $K(r)$ and a scalar random variable $r$ in the zeroth-order estimator, the gradient estimate can often be written in the form:

$$\hat{g_K} \approx \langle \nabla \mathcal{L}(\boldsymbol{\theta}), \boldsymbol{u} \rangle \cdot \boldsymbol{u} \cdot rK(r) \approx M \cdot rK(r), \qquad (21)$$

where $M$ approximates a constant related to the gradient magnitude and perturbation step size and $\boldsymbol{u}$ is a unit vector sampled uniformly at random. Under the formulation, we main focus is how the product $rK(r)$ affects the variance of the estimator. We know that variance is defined as:

$$\mathrm{Var}(X) = \mathbb{E}[X^2] - (\mathbb{E}[X])^2. \qquad (22)$$

If the estimator is approximately unbiased, i.e., $\mathbb{E}[\hat{g_K}] \approx c\nabla f(x)$ (c is a constant), we focus on $\mathbb{E}[\hat{g_K}^2]$.

In our estimator, the randomness comes from the term $rK(r)$. From the derivation above, we can rewrite the estimator in a simplified form:

$$\hat{g} = M \cdot (rK(r)). \qquad (23)$$

Then its second moment becomes:

$$\mathbb{E}[\hat{g_K}^2] \propto \mathbb{E}\left[(rK(r))^2\right]. \qquad (24)$$

Since $\boldsymbol{u}$ is a unit vector (or normalized), the fluctuation in the estimator is primarily greatly related to the term $rK(r)$. If $\mathbb{E}[(rK(r))^2]$ is large, then even with fixed $M$ and $\boldsymbol{u}$, the overall variance of the estimator will be large. Conversely, a smaller value of $\mathbb{E}[(rK(r))^2]$ helps reduce the estimator's variance. Since we use only three random perturbations to estimate the gradient, the variance can be relatively large in such low-perturbation settings. To address this, we constrain $r$ within the interval $[-1, 1]$, and gradually shrink the range of $r$ as the number of iterations increases. This strategy enables a trade-off between the variance and bias of the gradient estimation (Restricting the range of $r$ may result in gradient estimates that are not strictly unbiased; however, it effectively reduces the variance of the estimate when the number of perturbations is limited.).

### B.2   COMPLEMENTARY EXPERIMENTAL SETTINGS

**Baselines.** We select two representative baseline methods for our evaluation, i.e., MeZO (Malladi et al., 2023) and HiZOO (Zhao et al., 2024b). MeZO (Malladi et al., 2023) is a memory-efficient approach for LLM fine-tuning that avoids storing full perturbation vectors by regenerating them using a fixed seed. This eliminates the need for extra memory allocation and simplifies implementation. HiZOO (Zhao et al., 2024b) enhances convergence efficiency by incorporating approximate second-order curvature information into the optimization process, thereby addressing the typically slow convergence characteristic of first-order ZO methods.

**Hyperparameter settings.** Our experiments on RoBERTa-large, the OPT family, and LLaMA models adopt the hyperparameter configurations listed in Table 7. We also keep the hyperparameters consistent with MeZO, such as the learning rate and the perturbation scale. In our experiments, our kernel function constant $C$ is set to 4.

**Datasets.** In the RoBERTa-large experiments, we employ a range of classification benchmarks, including SST-2, SST-5, SNLI, TREC, MNLI, and RTE. These benchmarks cover a range of sentence-level and textual entailment tasks from previous NLP studies (Socher et al., 2013; Bowman et al., 2015; Voorhees & Tice, 2000; Yao & Lin, 2020; Dagan et al., 2005; Bar-Haim et al., 2006; Bentivogli

Table 7: The hyperparameters setting in our experiments.

| Experiment | Hyperparameters | Values |
|---|---|---|
| FT | Batch size | 8 |
| | Learning rate | {1e-5, 5e-5} |
| | Lr schedule | Constant for RoBERTa; Linear for OPT and LLaMA |
| MeZO | Batch size | {64, 16} |
| | Learning rate $\eta$ (Lr) | {1e-6, 5e-7} |
| | $\epsilon$ | 1e-3 |
| | Lr schedule | Constant for RoBERTa; Linear for OPT and LLaMA |
| MeZO LoRA | Batch size | {64, 16} |
| | Learning rate $\eta$ (Lr) | {1e-4, 5e-5} |
| | $\epsilon$ | 1e-2 |
| | Lr schedule | Constant for RoBERTa; Linear for OPT and LLaMA |
| KerZOO (LoRA) | Kernel function order $\beta$ | 3 |
| | $r$ | Shrink as iteration step increases |
| | Kernel function constant $C$ | 4 |

et al., 2009; Giampiccolo et al., 2007). To maintain consistency with prior works (Malladi et al., 2023; Zhao et al., 2024b), we set the test set size as 1000 examples. We examine both few-shot and many-shot regimes, setting the number of training examples per class to $k = 16$ or $k = 512$ and using the same number for validation. For experiments involving the OPT and LLaMA model families, we use the SuperGLUE benchmark (Wang et al., 2019), which includes tasks such as RTE (Dagan et al., 2005; Bar-Haim et al., 2006; Bentivogli et al., 2009; Giampiccolo et al., 2007), CB (De Marneffe et al., 2019), BoolQ (Clark & Lee, 2019), WIC (Pilehvar & Camacho-Collados, 2018), WSC (Levesque et al., 2012), and MultiRC (Khashabi et al., 2018). In addition, we include SST-2 (Socher et al., 2013) and two QA datasets, SQuAD (Rajpurkar et al., 2016) and DROP (Dua et al., 2019). For each dataset, we randomly select 1000 samples for training, 500 for validation, and 1000 for evaluation.

## B.3 MORE RESULTS

### B.3.1 MORE RESULTS ON LLaMA-3 MODEL

We conduct fine-tuning experiments of KerZOO on the LLaMA-3 model series. We use exactly the same hyperparameter settings as those used for the OPT family of models. The detailed results of the experiments are shown in Table 8 and 9 below.

Table 8: Experiment results on LLaMA3-3B (1000 training samples)

| Task | SST-2 | RTE | CB | WSC | WIC |
|---|---|---|---|---|---|
| FT | 94.2 | 81.2 | 91.4 | 72.2 | 63.8 |
| MeZO | 89.0 | 57.8 | 73.2 | **65.4** | 58.2 |
| KerZOO | **93.0** | **65.3** | **76.7** | **65.4** | **59.0** |

Table 9: Experiment results on LLaMA3-8B (1000 training samples)

| Task | SST-2 | RTE | CB | WSC | WIC |
|---|---|---|---|---|---|
| MeZO | 91.2 | 61.0 | 71.4 | 64.4 | 59.2 |
| KerZOO | **93.0** | **71.8** | **73.2** | **65.4** | **60.4** |

### B.3.2 MORE RESULTS ON LARGER MODEL

We conducted further experiments on larger model such as OPT-13B, and the results are presented in the table 10 below. We observe that the KerZOO method continues to outperform the MeZO and

HiZOO baselines in terms of performance while maintaining good time efficiency. For example, on the SQuAD dataset, our method improves time efficiency by 23% compared to MeZO, while also achieving an accuracy gain of approximately 4%.

Table 10: Experiment results on OPT-13B

| Method | SST2 | | RTE | | SQuAD | | WiC | | BoolQ | |
|---|---|---|---|---|---|---|---|---|---|---|
| | Acc | GPU hours | Acc | GPU hours | Acc | GPU hours | Acc | GPU hours | Acc | GPU hours |
| MeZO | 91.4 | 100% | 66.1 | 100% | 84.7 | 100% | 62.2 | 100% | 72.1 | 100% |
| HIZOO | 92.1 | 86% | 69.3 | 82% | 82.9 | 91% | 59.4 | 79% | 72.7 | 88% |
| KerZOO | **92.6** | 70% | **72.8** | 73% | **85.0** | 55% | **64.1** | 43% | **76.2** | 77% |

### B.3.3 MORE ANALYSIS ON MEMORY AND SPEED

Table 11: Memory and training time comparison of OPT-2.7B on SST-2 dataset (35 tokens per example on average)

| Method | Memory cost | Iteration step | GPU hours |
|---|---|---|---|
| FT | 45.4G | 9.3% | 16.8% |
| LoRA | 18.5G | 5.6% | 4.3% |
| MeZO | 10.7G | 100.0% | 100.0% |
| HiZOO | 11.3G | 59.2% | 87.4% |
| KerZOO | 14.7G | 25.0% | 76.2% |
| MeZO+LoRA | 5.5G | 74.1% | 43.7% |
| HiZOO+LoRA | 5.7G | 46.3% | 41.0% |
| KerZOO+LoRA | 5.7G | 16.3% | 29.5% |

Table 12: Memory and training time comparison of OPT-2.7B on RTE dataset (180 tokens per example on average)

| Method | Memory cost | Iteration step | GPU hours |
|---|---|---|---|
| FT | 62.2G | 10.0% | 16.2% |
| LoRA | 42.5G | 8.3% | 6.6% |
| MeZO | 13.5G | 100.0% | 100.0% |
| HiZOO | 13.8G | 63.3% | 88.9% |
| KerZOO | 14.5G | 22.5% | 72.3% |
| MeZO+LoRA | 7.5G | 73.3% | 34.8% |
| HiZOO+LoRA | 7.8G | 56.7% | 35.9% |
| KerZOO+LoRA | 7.7G | 7.6% | 11.5% |

We conduct experiments on the convergence and memory consumption of KerZOO on the OPT-2.7B model using the SST-2 and RTE datasets. As anticipated in our earlier analysis, KerZOO introduces only a slight increase in memory usage compared to the MeZO and HiZOO baselines. However, it exhibits fast convergence. For instance, on the RTE dataset, KerZOO combined with LoRA requires only about 11% of the training time required by MeZO under full fine-tuning, while still achieving competitive accuracy.

We show the training loss curves of the OPT-2.7B model on the SST-2 dataset in Figure 4. The comparison includes the original MeZO method with one perturbation per update (MeZO-1), MeZO with three perturbations (MeZO-3), and our proposed KerZOO method. Consistent with the findings reported in the original MeZO paper, we can observe that increasing the number of perturbations

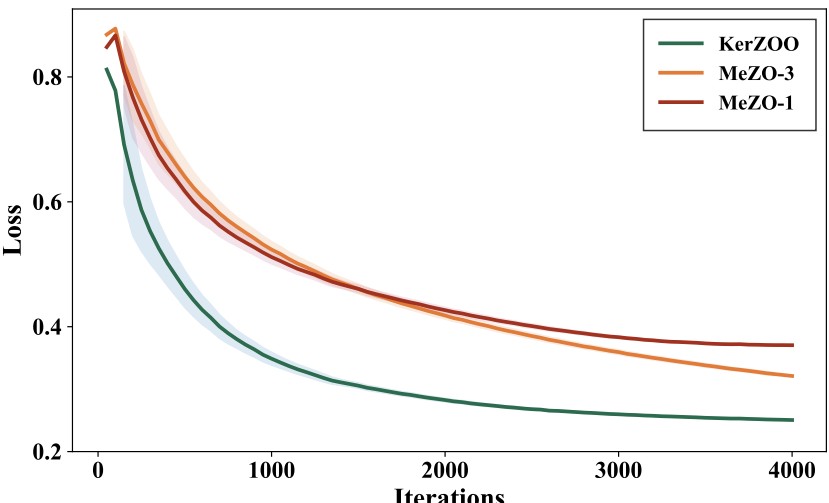

Figure 4: Training loss curves of MeZO with different perturbations and KerZOO

to three brings only marginal improvement in convergence. In contrast, our KerZOO demonstrates significantly faster convergence, highlighting the effectiveness of our kernel-based ZO optimization.

We also provide the plots of loss curves versus wall-clock times of fine-tuning OPT-2.7B on the SST-2 dataset with our method and MeZO (Table 13). When MeZO converges to the minimum loss of about 0.3392 at 1551s, KerZOO requires only 1182s to achieve the equivalent loss value, demonstrating faster convergence under the same training budget.

Table 13: Comparison of wall-clock time (in seconds) and loss between MeZO and KerZOO on OPT-2.7B with SST-2.

| #Iterations | 200 | 400 | 600 | 800 | 1000 | 1200 | 1500 | 2000 | 3000 | 4000 |
|---|---|---|---|---|---|---|---|---|---|---|
| Wall-clock time (s) for MeZO | 65 | 132 | 230 | 293 | 388 | 449 | 576 | 766 | 1172 | **1551** |
| Loss for MeZO | 0.7844 | 0.6701 | 0.5954 | 0.5422 | 0.5042 | 0.4732 | 0.4357 | 0.3933 | 0.3500 | **0.3392** |
| Wall-clock time (s) for KerZOO | 180 | 368 | 592 | 781 | 1012 | **1182** | 1518 | 2019 | 3063 | 4061 |
| Loss for KerZOO | 0.5939 | 0.4743 | 0.4104 | 0.3734 | 0.3497 | **0.3361** | 0.3149 | 0.2955 | 0.2801 | 0.2775 |

### B.3.4 ABLATION ABOUT THE $\epsilon$

We conduct experiments on convergence loss with various smaller $\epsilon$ (i.e., $0.5\epsilon_0$, $0.3\epsilon_0$, $0.1\epsilon_0$), and summarize the results in Table 14. For example, when $\epsilon = 0.5\epsilon_0$, the convergence loss increases to 0.3618 compared with 0.3392 under the original $\epsilon_0$, and becomes even higher when $\epsilon$ is further reduced. Therefore, although the bias are related to $\epsilon$ which is on the order of $\mathcal{O}(\epsilon^2)$, directly decreasing the $\epsilon$ is not a good choice. One possible explanation is that making $\epsilon$ extremely small introduces numerical instability in ZO estimation (Liu et al., 2024b; Jongeneel et al., 2024), which further highlights the effectiveness of our KerZOO method.

Table 14: Convergence loss with different $\epsilon$ values.

| Value of $\epsilon$ | Convergence loss |
|---|---|
| $\epsilon_0$ | 0.3392 |
| $0.5\epsilon_0$ | 0.3618 |
| $0.3\epsilon_0$ | 0.3865 |
| $0.1\epsilon_0$ | 0.4226 |

### B.3.5 COMBINING WITH QUANTIZATION

We also compare KerZOO to baselines with the same setting of 3 perturbations. Table 15 reports the results on SST-2 and SQuAD. Here, "-3" denotes experiments with 3 perturbations. In the table below, we also applied INT8 quantization to the intermediate variables $\Theta g$ in Algorithm 1 of KerZOO

(displayed as KerZOO* in the table above), following the method in (Shao et al., 2023), which further reduces KerZOO's memory overhead. For example, on the SST-2 dataset, the memory overhead decreases from 14.7G to 12.3G.

Table 15: Comparison of KerZOO with baselines under 3 perturbations on SST-2 and SQuAD.

| Method | SST2 Acc | SST2 GPU hours | SST2 memory | SQuAD Acc | SQuAD GPU hours | SQuAD memory |
|--------|----------|----------------|-------------|-----------|-----------------|--------------|
| MeZO-1 | 91.6 | 100% | 10.7G | 63.6 | 100% | 12.8G |
| MeZO-3 | 92.0 | 288% | 10.7G | 64.2 | 256% | 12.8G |
| HiZOO-1 | 90.8 | 88% | 11.8G | 66.0 | 92% | 14.1G |
| HiZOO-3 | 91.5 | 262% | 11.8G | 66.5 | 235% | 14.1G |
| KerZOO | 92.6 | 76% | 14.7G | 66.2 | 58% | 16.9G |
| KerZOO* | 91.8 | 81% | 12.3G | 66.2 | 64% | 14.6G |

### B.3.6 COMPARING WITH MORE BASELINES

We provide results of KerZOO on fine-tuning OPT-13B model compared with Sparse-MeZO and Subzero. For Sparse-MeZO (Liu et al., 2024c), on RTE, WiC, BoolQ, SST-2 and SQuAD with the OPT-13B model, we directly use the results from Table 2 in the SubZero paper. For SubZero (Yu et al., 2024), we reproduce the results using the same parameter settings as reported, and run experiments on five datasets with the OPT-13B model. The results are shown in Table 16.

Table 16: Comparison of KerZOO with Sparse-MeZO and SubZero on OPT-13B across five datasets.

| Datasets | SST2 | RTE | SQuAD | WiC | BoolQ |
|----------|------|-----|-------|-----|-------|
| Sparsezero | 92.3 | **76.9** | 77.9 | 58.2 | **76.5** |
| Subzero | 92.1 | 74.0 | 84.5 | 60.8 | 75.3 |
| KerZOO | **92.6** | 72.8 | **85.0** | **64.2** | 76.2 |

After reproducing SubZero, we found that its convergence efficiency is on par with, or even slightly worse than MeZO. For example, on the SST-2 dataset, SubZero's final converged loss is 0.3574, whereas MeZO's is approximately 0.3392. Our KerZOO achieves a convergence loss of 0.2764, which is the best.

### B.4 CONVERGENCE ANALYSIS

In this section, we give analysis on the convergence of KerZOO based on the former works on the kernel function (Bychkov et al., 2024; Lobanov & Gasnikov, 2023). We use $\|\boldsymbol{\theta}\|_p$ to denote the $\ell_p$-norm of the parameter $\boldsymbol{\theta}$ in LLMs, we use the notation $\|\boldsymbol{\theta}\|_2 = \|\boldsymbol{\theta}\|$ to express the Euclidean norm for simplicity. $\langle \boldsymbol{\theta}, \boldsymbol{\eta} \rangle := \sum_{k=1}^{d} \theta_i \eta_i$ is denoted as the standard inner product of $\boldsymbol{\theta}, \boldsymbol{\eta} \in \mathbb{R}^d$, where $\theta_i$ and $\eta_i$ are the $i$-th component of $\boldsymbol{\theta}$ and $\boldsymbol{\eta}$, respectively. $S_p^d(r) := \{\boldsymbol{\theta} \in \mathbb{R}^d : \|\boldsymbol{\theta}\|_p = r\}$ and $B_p^d(r) := \{\boldsymbol{\theta} \in \mathbb{R}^d : \|\boldsymbol{\theta}\|_p \leq r\}$ are used to express the sphere and the ball in the $\ell_p$-norm. Notation $\lesssim$ is used to denote the asymptotic inequality.

**Assumption 1 ($L$-smoothness).** Suppose loss function $\mathcal{L}(\boldsymbol{\theta})$ is an $L$-Lipschitz smooth function, or it has $L$-Lipschitz continuous gradient. If $\mathcal{L}(\boldsymbol{\theta})$ is continuously differentiable with respect to $\boldsymbol{\theta}$, and its gradient satisfies the Lipschitz condition for any $\xi$ and $\boldsymbol{\theta}, \boldsymbol{\eta} \in \mathbb{R}^d$

$$\|\nabla \mathcal{L}(\boldsymbol{\theta}) - \nabla \mathcal{L}(\boldsymbol{\eta})\| \leq L\|\boldsymbol{\theta} - \boldsymbol{\eta}\|. \tag{25}$$

**Assumption 2 (Higher-order smoothness).** Let $t$ denote the maximal integer which is strictly less than $\beta$. Let $\mathcal{F}_\beta(L)$ denote the set of all functions $\mathcal{L} : \mathbb{R}^d \to \mathbb{R}$ which can be derived for $t$ times and for any $\boldsymbol{\theta}, \boldsymbol{\eta} \in \mathbb{R}^d$ satisfy the Hölder-type condition:

$$\left| \mathcal{L}(\boldsymbol{\theta}) - \sum_{0 \leq |n| \leq t} \frac{1}{n!} D^n \mathcal{L}(\boldsymbol{\eta})(\boldsymbol{\theta} - \boldsymbol{\eta})^n \right| \leq L_\beta \|\boldsymbol{\theta} - \boldsymbol{\eta}\|, \tag{26}$$

where $L_\beta > 0$, the sum is over multi-index $n = (n_1, \ldots, n_d) \in \mathbb{N}^d$, the notation $n! = n_1! \cdots n_d!$, and

$$D^n \mathcal{L}(\boldsymbol{\eta}) \boldsymbol{v}^n = \frac{\partial^{|n|} \mathcal{L}(\boldsymbol{\eta})}{\partial \theta_1^{n_1} \cdots \partial \theta_d^{n_d}} v_1^{n_1} \cdots v_d^{n_d}, \tag{27}$$

where $|n| = n_1 + \cdots + n_d$, for all $\boldsymbol{v} = (v_1, \ldots, v_d) \in \mathbb{R}^d$.

Assumption 1 is a special case of Assumption 2 if $\beta = 2$.

Based on(Woodworth & Srebro, 2021), we have the following assumption:

**Assumption 3.** There exists $\sigma_*^2 \geq 0$ such that

$$\mathbb{E}\left[\|\mathbf{g}^* - \nabla \mathcal{L}(\boldsymbol{\theta}^*)\|^2\right] \leq \sigma_*^2. \tag{28}$$

which means that in overparameterized regime the variance of our estimation of stochastic gradient $\mathbf{g}^*$ at optimal point $\boldsymbol{\theta}^* = \arg\min_{\boldsymbol{\theta} \in \mathbb{R}^d} \mathcal{L}(\boldsymbol{\theta})$ can be upper bounded by using the minimum value $\mathcal{L}(\boldsymbol{\theta}^*)$. We will use this assumption for the gradient estimates.

Then by using two-point noisy zero-order oracle (Assumption 2), the gradient can be estimated as follows:

$$\mathbf{g}(\boldsymbol{\theta}, \boldsymbol{u}, r) = \frac{1}{2\epsilon} \left(\mathcal{L}(\boldsymbol{\theta} + \epsilon r \boldsymbol{u}) - \mathcal{L}(\boldsymbol{\theta} - \epsilon r \boldsymbol{u})\right) K(r) \boldsymbol{u}, \tag{29}$$

where $\epsilon > 0$ is a smoothing parameter (perturbation scale), $\boldsymbol{u} \in S_2^d(1)$ is a vector which is defined in Section 3, $K : [-1, 1] \to \mathbb{R}$ is a kernel function that satisfies

$$\mathbb{E}[K(a)] = 0, \quad \mathbb{E}[aK(a)] = C, \quad \mathbb{E}[|a||K(a)|] < \infty, \quad \text{and for all } j = 2, \ldots, t, \quad \mathbb{E}[a_j K(a)] = 0.$$

With Assumption 2, we will give a theorem for stating the convergence of KerZOO for LLM with two-point zeroth-order estimation. We denote:

$$\kappa_\beta = \int |a|^\beta |K(a)| \, da, \quad \text{and} \quad \kappa = \int |K(a)|^2 \, da.$$

**Theorem.** Let $\mathcal{L}(\cdot)$ satisfy Assumption 2 with parameter $\beta$. Let smoothing parameter be $\epsilon \leq \left(\frac{\psi}{\kappa_\beta L R}\right)^{1/(\beta-1)}$, then we use our Algorithm 1 to do gradient estimation. Let $\boldsymbol{\theta}_N^{ag}$ be the output of Algorithm 1, then

$$\mathbb{E}\left[\mathcal{L}(\boldsymbol{\theta}_N^{ag}) - \mathcal{L}(\boldsymbol{\theta}^*)\right] \leq \psi \tag{30}$$

in at most $N$ iterations and $T$ oracle calls. Note that in our case, $T$ equals to $N$ in our setting. $\rho$ denotes the radius to the optimal solution $\boldsymbol{\theta}^*$ such that $\|\boldsymbol{\theta}^* - \boldsymbol{\theta}_0\| \leq \rho$, and $\boldsymbol{\theta}_0$ is a starting point. Then, we have

$$N = T = \mathcal{O}\left(\max\left(\frac{L\rho^2}{\psi}, \frac{\kappa d \sigma_*^2 \rho^2}{\psi^2}\right)\right) \tag{31}$$

**Analysis.** Now we should find the upper bounds for smoothing parameter $\epsilon$, the asymptotic for iteration steps $N$ (number of oracle calls $T$). We consider to limit the bias and second moment of estimation of gradients. (1) For the bias of gradient approximation, it can be expressed as

$$b = \|\mathbb{E}\left[\mathbf{g}(\boldsymbol{\theta}_k, \boldsymbol{u}, r)\right] - \nabla \mathcal{L}(\boldsymbol{\theta}_k)\|. \tag{32}$$

$$b = \left\| \mathbb{E}\left[\frac{1}{2\epsilon}\left(\mathcal{L}(\boldsymbol{\theta}_k + \epsilon r \boldsymbol{u}) - \mathcal{L}(\boldsymbol{\theta}_k - \epsilon r \boldsymbol{u})\right) K(r)\boldsymbol{u}\right] - \nabla\mathcal{L}(\boldsymbol{\theta}_k) \right\|$$

$$\overset{\text{①}}{=} \left\| \mathbb{E}\left[\frac{1}{\epsilon}\left(\mathcal{L}(\boldsymbol{\theta}_k + \epsilon r \boldsymbol{u}) K(r)\boldsymbol{u}\right) - \nabla\mathcal{L}(\boldsymbol{\theta}_k)\right] \right\|$$

$$\overset{\text{②}}{=} \left\| \mathbb{E}\left[\frac{1}{\epsilon}\mathcal{L}(\boldsymbol{\theta}_k + \epsilon r \boldsymbol{u}) K(r)\boldsymbol{u}\right] - \nabla\mathcal{L}(\boldsymbol{\theta}_k) \right\|$$

$$\overset{\text{③}}{=} \left\| \mathbb{E}\left[(\nabla\mathcal{L}(\boldsymbol{\theta}_k + \epsilon r \boldsymbol{u}) r K(r)) - \nabla\mathcal{L}(\boldsymbol{\theta}_k)\right] \right\|$$

$$\overset{\text{④}}{\leq} \sup_{z \in S_2^d(1)} \mathbb{E}\left[(\nabla_z\mathcal{L}(\boldsymbol{\theta}_k + \epsilon r \boldsymbol{u}) - \nabla_z\mathcal{L}(\boldsymbol{\theta}_k)) r K(r)\right]$$

$$\overset{\text{⑤}}{\leq} \kappa_\beta \epsilon^{\beta-1} \frac{L}{(l-1)!} \mathbb{E}\left[\|\boldsymbol{a}\|^{\beta-1}\right] \leq \kappa_\beta \epsilon^{\beta-1} \frac{L}{(l-1)!} \frac{d}{d+\beta-1} \lesssim \kappa_\beta L \epsilon^{\beta-1} \tag{33}$$

where $\boldsymbol{a} \in B_2^d(1)$, ①: distribution of $\boldsymbol{u}$ is symmetric; ②: directly simplify the equation; ③: a version of the Stokes' theorem (Zorich & Paniagua, 2016) ; ④: gradient norm represent the supremum of directional derivatives $\nabla_z\mathcal{L}(\boldsymbol{\theta}) = \lim_{\epsilon\to 0}\frac{\mathcal{L}(\boldsymbol{\theta}+\epsilon z)-\mathcal{L}(\boldsymbol{\theta})}{\epsilon}$; ⑤ Taylor expansion.

(2) For the second moment of gradient approximation, it can be denoted as $\mathbb{E}\|\mathbf{g}(\boldsymbol{\theta}^*, \boldsymbol{u}, r)\|^2$. We have

$$\zeta^2 = \mathbb{E}\|\mathbf{g}(\boldsymbol{\theta}^*, \boldsymbol{u}, r)\|^2 = \mathbb{E}\left[\left\|\frac{1}{2\epsilon}\left(\mathcal{L}(\boldsymbol{\theta}^* + \epsilon r \boldsymbol{u}) - \mathcal{L}(\boldsymbol{\theta}^* - \epsilon r \boldsymbol{u})\right) K(r)\boldsymbol{u}\right\|^2\right]$$

$$= \frac{d}{4\epsilon^2}\mathbb{E}\left[\left(\mathcal{L}(\boldsymbol{\theta}^* + \epsilon r \boldsymbol{u}) - \mathcal{L}(\boldsymbol{\theta}^* - \epsilon r \boldsymbol{u})\right)^2 (K(r))^2\right]$$

$$\overset{\text{①}}{\leq} \frac{\kappa d}{2\epsilon^2}\mathbb{E}\left[\left(\mathcal{L}(\boldsymbol{\theta}^* + \epsilon r \boldsymbol{u}) - \mathcal{L}(\boldsymbol{\theta}^* - \epsilon r \boldsymbol{u})\right)^2\right]$$

$$\overset{\text{②}}{\leq} \frac{\kappa d}{2\epsilon^2}\left(\epsilon^2\mathbb{E}\left[\|\nabla\mathcal{L}(\boldsymbol{\theta}^* + \epsilon r \boldsymbol{u}) + \nabla\mathcal{L}(\boldsymbol{\theta}^* - \epsilon r \boldsymbol{u})\|^2\right]\right)$$

$$= \frac{\kappa d}{2\epsilon^2}\left(\epsilon^2\mathbb{E}\left[\|\nabla\mathcal{L}(\boldsymbol{\theta}^* + \epsilon r \boldsymbol{u}) + \nabla\mathcal{L}(\boldsymbol{\theta}^* - \epsilon r \boldsymbol{u}) \pm 2\nabla\mathcal{L}(\boldsymbol{\theta}^*)\|^2\right]\right)$$

$$\overset{\text{③}}{\leq} 4d\kappa\|\nabla\mathcal{L}(\boldsymbol{\theta}^*)\|^2 + 4d\kappa L^2\epsilon^2\mathbb{E}[\|\boldsymbol{u}\|^2]$$

$$\overset{\text{④}}{\leq} 4\kappa d\sigma_*^2 + 4\kappa d L^2\epsilon^2\mathbb{E}[\|\boldsymbol{u}\|^2] = 4\kappa d\sigma_*^2 + 4\kappa d^2 L^2\epsilon^2 \tag{34}$$

where ① Inequality of squared norm of the sum, inequality between positive random variables and independence of the noise; ② Wirtinger–Poincaré inequality; ③ $L$-smoothness of $\mathcal{L}$; ④ Overparameterization assumption.

**Sketchup for covergence.** According to (Woodworth & Srebro, 2021), we can replace the following expression of the bound for biased oracle with the bias and second moment estimation above. And we have

$$\mathbb{E}[\mathcal{L}(\boldsymbol{\theta}_N^{ag}) - \mathcal{L}(\boldsymbol{\theta}^*)] = c\left(\frac{L\rho^2}{N^2} + \frac{L\rho^2}{N} + \frac{\zeta\rho}{\sqrt{N}} + b\rho + \frac{b^2}{2L}N\right)$$

$$\lesssim \frac{L\rho^2}{N^2} + \frac{L\rho^2}{N} + \frac{1}{\sqrt{N}}\left(\sqrt{\kappa d\sigma_*^2} + \sqrt{\kappa d^2 L^2 \epsilon^2}\right)\rho$$

$$+ \left(\kappa_\beta L\epsilon^{\beta-1}\right)\rho + \frac{1}{L}\left(\kappa_\beta L\epsilon^{\beta-1}\right)^2 N$$

$$\leq \frac{L\rho^2}{N^2} + \frac{L\rho^2}{N} + \frac{\sqrt{\kappa d\sigma_*^2}\rho}{\sqrt{N}} + \frac{\sqrt{\kappa d}L\epsilon\rho}{\sqrt{N}}$$

$$+ \kappa_\beta L\epsilon^{\beta-1}\rho + \kappa_\beta^2 L\epsilon^{2(\beta-1)}N \tag{35}$$

We use $①$ $\frac{L\rho^2}{N^2}$, $②$ $\frac{L\rho^2}{N}$, $③$ $\frac{\sqrt{\kappa d}\sigma_*\rho}{\sqrt{N}}$, $④$ $\frac{\sqrt{\kappa d}L\epsilon\rho}{\sqrt{N}}$, $⑤$ $\kappa_\beta L\epsilon^{\beta-1}\rho$, $⑥$ $\kappa_\beta^2 L\epsilon^{2(\beta-1)}N$, to denote every term in the expression. Then we try to use $\psi$ to bound every term above and now we attempt to find all parameter values for all cases.

$$\mathbb{E}[\mathcal{L}(\boldsymbol{\theta}_N^{ag}) - \mathcal{L}(\boldsymbol{\theta}^*)] = \frac{L\rho^2}{N^2} + \frac{L\rho^2}{N} + \frac{\sqrt{\kappa d}\sigma_*\rho}{\sqrt{N}} + \frac{\sqrt{\kappa d}L\epsilon\rho}{\sqrt{N}} + \kappa_\beta L\epsilon^{\beta-1}\rho + \kappa_\beta^2 L\epsilon^{2(\beta-1)}N \leq 6\psi. \tag{36}$$

For $①$, $②$, and $③$, we have $N$:

$$\frac{L\rho^2}{N^2} \leq \psi \quad \Longrightarrow \quad N \geq \sqrt{\frac{L\rho^2}{\psi}} = \mathcal{O}\left(\sqrt{\frac{L\rho^2}{\psi}}\right)$$

$$\frac{L\rho^2}{N} \leq \psi \quad \Longrightarrow \quad N \geq \frac{L\rho^2}{\psi}$$

$$\frac{\sqrt{\kappa d}\sigma_*\rho}{\sqrt{N}} \leq \psi \quad \Longrightarrow \quad N \geq \frac{\kappa d\sigma_*^2\rho^2}{\psi^2} \tag{37}$$

Note that $\frac{L\rho^2}{\psi} > \sqrt{\frac{L\rho^2}{\psi}}$. That leads to the following expression for $N$

$$N = T = \mathcal{O}\left(\max\left(\frac{L\rho^2}{\psi}, \frac{\kappa d\sigma_*^2\rho^2}{\psi^2}\right)\right). \tag{38}$$

For $④$, $⑤$, and $⑥$, we can have $\epsilon$:

$$\frac{\sqrt{\kappa d}L\epsilon\rho}{\sqrt{N}} \leq \psi \quad \Longrightarrow \quad \epsilon \leq \frac{\sqrt{N}\psi}{\sqrt{\kappa d}L\rho} = \max\left(\frac{\sqrt{L\rho^2/\psi}\cdot\psi}{\sqrt{\kappa d}L\rho}, \frac{\sqrt{\kappa d\sigma_*^2\rho^2/\psi^2}\cdot\psi}{\sqrt{\kappa d}L\rho}\right)$$

$$= \max\left(\frac{\psi^{1/2}}{\sqrt{\kappa}L^{1/2}}, \frac{\sigma_*}{\sqrt{d}L}\right)$$

$$\kappa_\beta^2 L\epsilon^{2(\beta-1)}N \leq \psi \quad \Longrightarrow \quad \epsilon \leq \left(\frac{\psi}{\kappa_\beta^2 LN}\right)^{\frac{1}{2(\beta-1)}}$$

$$\kappa_\beta L\epsilon^{\beta-1}\rho \leq \psi \quad \Longrightarrow \quad \epsilon \leq \left(\frac{\psi}{\kappa_\beta L\rho}\right)^{\frac{1}{\beta-1}} \tag{39}$$

$$\epsilon = \min\left(\max\left(\frac{\psi^{1/2}}{\sqrt{\kappa}L^{1/2}}, \frac{\sigma_*}{L}\right), \left(\frac{\psi}{\kappa_\beta^2 LN}\right)^{\frac{1}{2(\beta-1)}}, \left(\frac{\psi}{\kappa_\beta L\rho}\right)^{\frac{1}{\beta-1}}\right) = \left(\frac{\psi}{\kappa_\beta L\rho}\right)^{\frac{1}{\beta-1}}. \tag{40}$$

