# OpenReview forum: "KerZOO: Kernel Function Informed Zeroth-Order Optimization for Accurate and Accelerated LLM Fine-Tuning"
_ICLR.cc/2026/Conference — ICLR 2026 Conference Desk Rejected Submission_

### Official Review · Reviewer_HQgz · 2025-10-27

**Soundness:** 3
**Presentation:** 1
**Contribution:** 3
**Rating:** 4
**Confidence:** 3

**Summary:**

This paper introduces KerZOO, a kernel function-based zeroth-order (ZO) optimization framework for efficient fine-tuning of large language models (LLMs). KerZOO uses a Taylor expansion to identify the lower-order bias in ZO gradient estimation and introduces a kernel function satisfying specific moment conditions to eliminate this bias. Extensive empirical results show higher accuracy and faster convergence of KerZOO compared with existing baselines across multiple benchmarks.

**Strengths:**

The paper makes an interesting observation that the lower-order bias in ZO gradient estimation can be removed by incorporating an additional scalar random variable $r$ and applying a kernel function weighting. This approach is promising and has the potential to address the slow convergence commonly observed in existing ZO optimization methods.

**Weaknesses:**

The paper is not clearly written. There is limited discussion or intuition provided to explain the proposed kernel function and rationale behind Algorithm 1. Some experimental results are also difficult to interpret (e.g., Tables 4, 5, and 6). Overall, the writing needs to be significantly improved to meet publication standards. Additionally, the effect of kernel weighting on variance should be explored in more depth.

**Questions:**

- In Section 3.4, why is $K_\beta(r)$ chosen in the specific form given in Eq (17)? The proposed kernel function seems to appear from nowhere. A discussion clarifying the reasoning behind this particular choice would help readers better understand the kernel design (also $K_\beta(r)$ and $K(r)$ in Eq (17) appear redundant, remove one).

- It would be helpful to discuss whether there exist other kernel functions satisfying the moment conditions and how to compare different design choices. Are there any general strategies for constructing the kernels satisfying the moment conditions?

- KerZOO removes the lower-order bias, but it introduces another random variable $r$ which could increase the estimator's variance. The paper briefly discusses variance in the appendix, but the explanation is very heuristic. Can the authors provide a more rigorous analysis of how the kernel design influences variance, or at least empirical evidence showing that variance does not increase significantly?

- The constant $C$ seems to be an important hyperparameter in the kernel design, but it is not clearly discussed. Table 6 gives some experimental results for different $C$ values, but it gives no intuition or guidance on how to select it. Since $C$ may affect higher-order bias and possibly variance, the paper should discuss the theoretical and/or practical guidances for choosing $C$.

- Algorithm 1 uses an exponential moving average at every iteration, but the paper gives no explanation or motivation for this algorithm choice. Since Algorithm 1 is presented as the main fine-tuning procedure, a detailed discussion is needed. Also $\theta_t^{md}$ doesn't seem to be used in Algorithm 1?

- What does "percentage" represent in Table 4, and what does "loss at iteration" mean in Tables 5 and 6?

Minor comments:

- Line 128: perturbation. (period)

- Eq (4): replace $\nabla^2 f(x)$ with $\nabla^2 \mathcal{L}(\theta)$.

- Change the numbering of Figure 3 and Figure 2?

---

> ### Author Response · Authors · 2025-11-26
>
> **[About the kernel function]**
>
> Thanks to the reviewer for this comment.
> We use Legendre polynomials as a tool to construct the kernel function. The general process is as follows:
>
> Construction of the kernel function $k_\beta(r)$ can be summarized as follows:
>
> 1. Choice of orthonormal polynomial basis
> Let $\{p_m(r)\}$ be the orthonormal polynomials w.r.t. $r\sim U([-1,1])$, e.g.
> $$
> p_m(r)=\sqrt{2m+1}L_m(r),
> $$
> where $L_m$ is the $m$‑th Legendre polynomial.
>
> 2. Linear combination via derivatives at zero
> Define
> $$
> k_\beta(r)
> = C\sum_{m=0}^\beta p_m'(0)p_m(r).
> $$
> Then for $s=0,2,\dots,\beta$, we want to:
> $$
> \int_{-1}^1 r^sk_\beta(r)dr = 0,
> \quad
> \int_{-1}^1 rk_\beta(r)dr = 2C.
> $$
> Equivalently, since $r\sim U([-1,1])$ has density $p(r)=\tfrac12$,  and we want to satisfy the higher  order moment and first order moment:
> $$
> \mathbb{E}[r^sk_\beta(r)]
> =\int_{-1}^1 r^sk_\beta(r)p(r)dr
> =\frac12\int_{-1}^1 r^sk_\beta(r)dr
> =0,
> \quad
> \mathbb{E}[rk_\beta(r)]
> =\frac12\int_{-1}^1 rk_\beta(r)dr
> =C.
> $$
>
>
> 3. Explicit closed‑form expressions
> Computing $p_m'(0)$ for normalized Legendre polynomials yields:
> - **$\beta=1$:**
>   $$
>   k(r)=3Cr.
>   $$
> - **$\beta=3$:**
>   $$
>   k(r)=\frac{15}{4}Cr(5-7r^2).
>   $$
> - **$\beta=5$:**
>   $$
>   k(r)=\frac{195}{64}Cr(99r^4 - 126r^2 + 35).
>   $$
>
>
> These $k_\beta(r)$ satisfy all moment conditions up to order $\beta$, eliminate all Taylor errors except the linear term, and yield an $O(\delta^{\beta+1})$ bias with unbiased gradient estimates. For example, we use the kernel function of  $\beta=3$:
> $$
> \mathbb{E}[r^3\frac{15}{4}Cr(5-7r^2)]
> =\int_{-1}^1 r^3\frac{15}{4}Cr(5-7r^2)p(r)dr
> =\frac12\int_{-1}^1 r^3\frac{15}{4}Cr(5-7r^2)dr
> =0,
> \quad
> \mathbb{E}[r\frac{15}{4}Cr(5-7r^2)]
> =\frac12\int_{-1}^1 r\frac{15}{4}Cr(5-7r^2)dr
> =C.
> $$
>
>
>
> More explanation: Why using Legendre polynomials: the literature we have cited in our paper [1, 2, 3, 4] indicates that constructing these kernels with Legendre polynomials provides several mathematical benefits: Decoupled Orders via Orthogonality: The natural orthogonality of Legendre polynomials ensures that when they are linearly combined to cancel out moments up to order β, the contributions from different orders remain decoupled and do not affect one another. Optimal Degree for Moment Cancellation: Nullifying all moments from order 0 to β on a symmetric interval requires a polynomial of at least degree β. The Legendre polynomials form a consistent hierarchy where the m-th polynomial, Pm​(x), is of degree m. This structure allows us to construct a kernel of order β without using any unnecessarily high-degree terms, thus yielding a construction with the minimum possible polynomial degree. Closed-Form Coefficients without Solving Linear Systems: Due to the orthogonality, the kernel's coefficients can be expressed analytically. One can write down the complete kernel formula simply by knowing the derivative of the Legendre polynomials at the origin, Pm′​(0). The coefficients are available in a closed form, which eliminates the need to numerically solve a (β+1)×(β+1) system of linear equations. This approach is both more convenient and circumvents potential numerical instability.
>
> [1] Bach, Francis, and Vianney Perchet. "Highly-smooth zero-th order online optimization." Conference on Learning Theory. PMLR, 2016.
>
> [2] Bychkov, Georgii, et al. "Accelerated zero-order SGD under high-order smoothness and overparameterized regime." arXiv preprint arXiv:2411.13999 (2024).
>
> [3] Nevai, Paul G. Orthogonal polynomials. Vol. 213. American Mathematical Soc., 1979.
>
> [4] Gautschi, Walter. Orthogonal polynomials: computation and approximation. OUP Oxford, 2004.
>
> **[Different kernel functions]**
>
> Thanks to the reviewer for this comment.
> Our kernel $K_{\beta}(r)$ corresponds to the Legendre polynomial family that satisfies the required moment conditions.  Different values of $\beta$ produce different valid kernels within this family, and all of them are admissible choices for KerZOO. They are all from the construction of Legendre polynomials.
>
> We choose $\beta = 3$ because it gives the lowest-degree kernel that cancels the lower-order bias while keeping variance small and it is good enough for convergence.  Table 5 provides an ablation over different $\beta$ values, showing that the performance is extremely stable and that $\beta = 3$ is already sufficient to achieve strong accuracy.

---

> ### Author Response · Authors · 2025-11-26
>
> **[Explanation of estimator's variance]**
>
> Thanks very much for the reviewer’s comment.
>
> We understand the reviewer’s concern about whether introducing the additional scalar $r$ may increase variance.
> As explained in Appendix B.1, our design explicitly avoids this issue: during training, we gradually shrink the support of $r$, which naturally drives all $r$-dependent moments toward zero. Consequently, the variance term associated with $r$ also decreases toward zero as training progresses.
>
> Importantly, this shrinking process does not negatively affect convergence. For example, in RoBERTa-large trained for 100k iterations (Figure 1), the early stage where most of the loss reduction happens, shows almost no change in $r$, meaning the moment conditions remain essentially satisfied and the estimator behaves like its ideal theoretical form.  In the late stage, as $r$ gradually decreases, both the variance and the moment terms approach zero, which stabilizes the loss curve and prevents oscillation.
>
> To further verify this effect, we additionally conducted experiments comparing the cases where r is gradually reduced versus kept fixed. On the SST-2 dataset with the OPT-2.7B model, the convergence curves of the two settings almost overlap in the early stage of training: both reach a loss of approximately 0.34 which is comparable to MeZO’s loss after 4,000 steps within only about 1100-1200 steps. However, in the mid-to-late training stage, the curves diverge. When r is kept fixed, the loss begins to rise, reaching 0.42 at 2,000 steps and further increasing to 0.53 at 4,000 steps. In contrast, when r is gradually reduced, the loss continues to decrease and reaches a significantly lower value of 0.27 at 4,000 steps. These results indicate that shrinking r helps maintain training stability in the later stages, while having little to no negative effect on early-stage convergence.
>
> **[Guidance for choosing C]**
>
> Thanks for the reviewer's comment. We agree that $C$ is an important hyperparameter of the kernel.   Table 6 already provides sensitivity experiments across a wide range of $C$ values. From Equation (16), we can see that the magnitude of $C$ is closely tied to the scale of the ZO gradient estimator.  When designing the ablation study for $C$, our initial hypothesis was that a too-small $C$ would cause the estimated gradient to vanish, whereas an excessively large $C$ would produce overly large gradient updates.  Therefore, we began our ablation or process of choosing C by setting $C$ to values around 5. In our experiments, we find that $C = 4$ consistently yields the best performance, and this observation holds across all evaluated models.
>
> **[Unclarity of notation]**
>
> Thank you for pointing this out. I am very sorry that our notation was inconsistent: $\theta_t^{md}$ should indeed be used in Step 6 of Algorithm 1, and this will be corrected in the final version. The exponential moving average included in the algorithm follows the standard smoothing technique used in prior work [1]. It does not affect the kernel function, the theoretical bias cancellation, or any of our experimental results.
>
> [1] Accelerated Zero-Order SGD under High-Order Smoothness and Overparameterized Regime (arXiv:2411.13999).
>
> **[Explanation of Table caption]**
>
> Thanks very much for reviewer's comment.
> In Table 4, “percentage” means the proportion of total training time relative to the MeZO baseline, which is taken as 100%. We compare how many percent of MeZO’s total GPU hours are required for KerZOO and other methods to reach the same convergence target. “Loss at iteration” in Tables 5 and 6 denotes the training loss achieved at the corresponding fixed number of iterations shown on the left (#Iteration). This metric ensures that all hyperparameter settings are evaluated under the same iteration budget and therefore the same compute. We will clarify these definitions directly in the table captions.

---

### Official Review · Reviewer_5uGb · 2025-11-01

**Soundness:** 3
**Presentation:** 2
**Contribution:** 2
**Rating:** 2
**Confidence:** 5

**Summary:**

The paper proposes **KerZOO**, a kernel-guided zeroth-order (ZO) optimization method for LLM fine-tuning. It analyzes the low-order bias introduced by random perturbations in ZO gradient estimation and designs kernels satisfying moment conditions (e.g., $\mathbb{E}[rK(r)] = C$, $ \mathbb{E}[r^3K(r)] = 0 )$ to cancel the dominant bias term. The resulting estimator is simple to implement (few directions, e.g., $n=3$ and drops into standard ZO-SGD. Experiments on RoBERTa-large, OPT-2.7B/6.7B, and LLaMA-3 3B/8B across classification and generation tasks show faster convergence and competitive or better accuracy than MeZO/HiZOO, with sizable reductions in GPU hours; the method remains compatible with LoRA. Ablations on kernel order $(\beta)$ and scaling constant $C$ support the design choices.

**Strengths:**

* Precisely targets ZO’s low-order bias and provides principled kernel conditions with expectation-level analysis.
* Kernel weighting and scalar perturbations integrate cleanly; works with very few directions $(n  \approx 3)$.
* Substantial cuts in training steps/GPU hours while maintaining or improving accuracy.
* Results span multiple model families/sizes and tasks, under both full-finetuning and PEFT (LoRA).
* Sensitivity to $\beta$ and $C$, plus memory/time comparisons, aid reproducibility and deployment.

**Weaknesses:**

1. **Near-duplicate figures/tables.** The paper’s plotting/table template and ordering are *highly similar* to submission **#12350**, with only color changes: **#12282 Fig.1 / Fig.2 / Fig.3 ≈ #12350 Fig.3 / Fig.2 / Fig.4** (same axes styles, legend shapes, and layout).

2. **If a shared template is acceptable, how do you explain drifting baselines?** Under ostensibly comparable settings, baselines differ across the two papers in ways that **systematically favor each paper’s own method**.

   * Example (**Table 2 · DROP**): **#12282** MeZO-LoRA=13.4 / HiZOO-LoRA=13.9 / KerZOO-LoRA=14.7 **vs.** **#12350** MeZO-LoRA=19.2 / HiZOO-LoRA=18.3 / P-GAP-LoRA=22.5.
   * Example (**Table 3**): the same cross-paper baseline discrepancy appears **column-wise** under matched configurations.

3. **Even within this paper, baselines are selectively changed without explanation.** In **Table 1**, the **Zero-shot** row mirrors the MeZO paper, but **MeZO-LoRA** departs markedly. The paper does not clarify which baselines are re-runs vs. prior quotes, nor why numbers were altered—changes consistently benefit the proposed method.

4. **Reproducibility gap.** The released code lacks **requirements.txt / environment specs, hyperparameters/seeds, making the results non-reproducible and preventing verification of the baseline inconsistencies.

**Questions:**

See Weaknesses.

---

> ### Author Response · Authors · 2025-11-28
>
> Thanks very much for reviewer's comments.
> In our implementation of P-GAP, we observed that P-GAP achieves convergence very early during training, especially on generative datasets. In contrast, on the DROP dataset, MeZO+LoRA and HiZOO+LoRA baselines reach only 13.4 and 13.9 accuracy even after 4,000 training steps. While we extended the baseline training to about 12,000 steps, which finally matches the 19.2 and 18.3 accuracy reported as the MeZO+LoRA and HiZOO+LoRA baseline in the P-GAP paper. However, P-GAP itself reaches approximately 22.5  accuracy within only around 1200 steps.
> In addition, all KerZOO experiments (including baselines) were conducted on both A100 and A6000 GPUs, whereas all P-GAP experiments were run only on A100 GPUs. For consistency and fairness, we independently reran the baselines for both papers.
>
> For Table 1, all zeroth-order baselines,  including MeZO full fine-tuning and MeZO-LoRA, were re-run by us to ensure consistency. For baselines unrelated to zeroth-order optimization, such as the zero-shot results, we directly adopt the numbers reported in the MeZO paper.

---

### Official Review · Reviewer_v2NT · 2025-11-01

**Soundness:** 2
**Presentation:** 2
**Contribution:** 3
**Rating:** 4
**Confidence:** 3

**Summary:**

This paper proposes KerZOO, a kernel-function–informed zeroth-order (ZO) optimizer for LLM fine-tuning. The core idea is to attach a random scalar r to each two-point ZO query and weight it with a kernel K(r) whose moments enforce $E[rK(r)]=C$ and $E[r^{3}K(r)]=0$, thereby canceling the leading $O(\epsilon^{2})$ bias term in the ZO gradient estimator and leaving only higher-order bias. Algorithm 1 implements this estimator with multiple perturbation directions per step and uniform $r\sim[-1,1]$ sampling. Experiments on RoBERTa-large, OPT, and LLaMA report faster convergence and accuracy gains over MeZO/HiZOO, with large reductions in normalized GPU hours (e.g., up to 74% on WSC) at comparable memory cost. Overall, the paper targets a concrete weakness of ZO (biased estimates) with a clear analytic prescription (kernel moment conditions) and backs it with broad empirical evidence.

**Strengths:**

- Clear and targeted theory: the derivation isolates the lower-order bias in two-point ZO and gives explicit kernel moment conditions ($E[rK(r)]=C$, $E[r^{3}K(r)]=0$) that remove it in expectation, which is simple to check/implement.
- Practical algorithmic wrapper: KerZOO fits the standard two-point estimator with minimal changes (draw $u$, draw $r$, apply $K(r)$) and uses a small number of directions (default $n=3$), making adoption straightforward.
- Broad empirical wins with efficiency: consistent improvements over MeZO/HiZOO across GLUE/SuperGLUE-style tasks and models, plus sizable reductions in training iterations/GPU hours at similar memory footprints (e.g., Tables/Figs for RoBERTa/OPT, and the SQuAD time-memory table).

**Weaknesses:**

- Theoretical clarity/notation: the Taylor expansions mix notations (e.g., $∇^{2}f(x)$ vs. $∇^{2}L(θ)$) and use $D^{3}∇L$ (which suggests a 4-th order derivative) without a clean justification; please tighten the derivation around Eqs. (4)–(8) and state smoothness/independence assumptions precisely.
- Assumption–implementation gap: the theory relies on $r \in [-1,1]$ with moment constraints, but the method later “shrinks” the range of $r$ over training to reduce variance, breaking $E[rK(r)]=C$ and $E[r^{3}K(r)]=0$; the paper acknowledges this qualitatively but lacks a bias bound under shrinking-$r$. The convergence theorem also hides constants ($\kappa,\kappa_\beta$) and leaves the practical regime unclear.
- Fairness and accounting: core results set KerZOO to $n=3$ directions by default, but it is not always explicit whether MeZO/HiZOO use the same number of directions or the same forward-pass budget; key tables highlight “iterations/GPU hours” but not queries per update. Although Table 15 adds a 3-direction ablation, the main tables would benefit from strict normalization by total forward evaluations. Also, Algorithm 1’s $\beta_t$ schedule is ambiguous (typo?), complicating reproduction.

**Questions:**

- Evaluation protocol & reporting: For Tables 1–3, please match MeZO/HiZOO to the same number of perturbation directions (e.g., n=3) and identical forward-pass budgets, so that results are fully reproducible.
- Shrinking-$r$ schedule: since you shrink the support of $r$ during training, what is the resulting residual bias term (in place of $E[r^{3}K(r)]=0$)? Can you provide a bound that depends on the current interval and show convergence with this controlled bias? Ideally add a figure/table showing accuracy and loss vs. shrink schedule.
- Budget normalization & hyperparameters: for Tables 1–3, can you report results where each method is matched on total forward-pass queries (or function evaluations), and clarify the exact perturbation count for MeZO/HiZOO in those tables? Also specify the intended $\beta_t$ formula in Algorithm 1 and provide sensitivity to $β$, $C$ (you use $C=4$), and $n$.

---

> ### Author Response · Authors · 2025-11-26
>
> **[Clarity of notations]**
>
> We thank the reviewer for pointing this out.
> The Taylor expansion and differential operators in Eqs. (4–8) are standard second- and third-order derivatives; the notation ∇²f vs. ∇²L and D³∇L will be unified and clarified in the final version to avoid confusion.
>
> **[Shrinking of r]**
>
> Thank you for the insightful comment.
>
> We clarify that the purpose of the shrink-$r$ schedule is *not* to change the theoretical bias cancellation, but to ensure stable training when we match MeZO’s iteration budget (e.g., 100k steps). Our convergence analysis shows that KerZOO can converge in a small number of iterations, but in practice we must run the *same number of update steps* as the baseline for a fair comparison. Thus, after KerZOO has already converged, the remaining iterations should not introduce additional variance that would destabilize the loss curve and this is exactly what shrink-$r$ is designed to address.
>
> During the early stage of training, where convergence actually happens and ZO updates dominate model improvement, the values of $r$ remain extremely close to 1. For example, on SNLI with 100k steps on RoBERTa-large, we use:
>
> $$
> r = 1 - \frac{t}{100000}.
> $$
>
> In the early phase (the first 10–20% of training), this means $r \approx 1$, and therefore the moment conditions
>
> $$
> \mathbb{E}[\ r K(r) \] = C, \qquad
> \mathbb{E}[\ r^3 K(r) \] = 0
> $$
>
> remain essentially unchanged. Thus, the bias-cancellation effect holds precisely during the crucial period in which KerZOO achieves rapid descent. Indeed, Figure 1 shows that KerZOO reaches its low-loss region far earlier than MeZO, confirming that shrink-$r$ does not interfere with convergence.
>
> In the mid- and late-training phases, after KerZOO has already reached its converged region, shrinking $r$ gradually lowers the variance of the estimator, as described in Appendix B.1.
> As $r_t \to 0$, both moment terms smoothly decay:
>
> $$
> \mathbb{E}[\ r K(r) \] \to 0, \qquad
> \mathbb{E}[\ r^3 K(r) \] \to 0,
> $$
>
> and the estimator transitions into a low-variance regime. This reduces oscillations in the tail of training and keeps the loss curve flat and stable, without introducing a new source of bias. This behavior is exactly what we observe in Figure 1 for RoBERTa-large.
>
> Finally, regarding the constants in the convergence theorem: as shown in Appendix B.4, they are defined as
>
> $$
> \kappa_{\beta} = \int |a|^{\beta} |K(a)| \, da, \qquad
> \kappa = \int |K(a)|^{2} \, da,
> $$
>
> and depend *only* on the kernel choice, not on the model architecture, data distribution, or parameter dimension.
> For our kernel (β = 3, C = 4), both values are finite, small, and fixed throughout training, so they do not restrict the practical regime or introduce instability.
>
> We will make these points clearer in the revision.
>
> **[About setting baseline to same perturbations]**
>
> Thanks very much for reviewer's comment.
> We respectfully argue that normalizing all methods to the same number of perturbation directions is not a meaningful fairness criterion for KerZOO. The motivation and goal of KerZOO is precisely to reduce the higher order bias so that fewer training time is needed to achieve faster convergence. Baselines do not benefit from additional directions; in fact, increasing their direction count only increases GPU cost without improving convergence (as shown in Figure 4). Normalizing all methods to use the same n would remove the efficiency advantage of KerZOO comparing to baselines (because baselines need more training time than KerZOO even n=1). KerZOO is explicitly designed to reduce estimator bias so that fewer training times are sufficient to reach the same or better accuracy. Therefore, matching “directions per update” is not the right fairness criterion or the motivation of KerZOO: the core contribution of KerZOO is precisely that for the same or less total training time, it achieves higher accuracy and faster loss reduction compared to MeZO/HiZOO. Our goal is not to compare accuracy under artificially equalized forward-pass counts.
> Therefore, normalizing perturbation counts does not lead to a fair comparison in terms of training efficiency.

---

> ### Author Response · Authors · 2025-11-26
>
> **[Shrinking r strategy]**
>
> Thanks to the reviewer for the comment.
> Early training: negligible effect on the bias
> On SST2 with 100000 steps, the schedule
> $$
> r = 1 - \frac{step}{20000}
> $$
> keeps $r$ extremely close to 1 during the early training. In this regime, the moment conditions
> $$
> \mathbb{E}[rK(r)] = C, \qquad
> \mathbb{E}[r^3 K(r)] = 0
> $$
> remain essentially unchanged, so the original $O(\epsilon^4)$ bias cancellation continues to hold throughout the entire descent region of training. This matches what Figure 1 shows: KerZOO’s loss curve decreases normally and even descends even faster than MeZO, confirming the negligible impact of shrink-$r$.
>
> Late training: both the bias term and the moment conditions approach zero after convergence
> When the loss has already plateaued and $r$ becomes small, both kernel moments naturally satisfy
>
> $$
> \mathbb{E}[rK(r)] \to 0, \qquad
> \mathbb{E}[r^3 K(r)] \to 0.
> $$
>
> Therefore, the residual bias term
>
> $$
> O(\epsilon^4) +O\left( \mathbb{E}[\ r K(r) \] + \mathbb{E}[\ r^3 K(r) \] \right) = O(\epsilon^4) + o(1)
> $$
>
>
>
> which vanishes as $r \to 0$. This yields a controlled, diminishing bias, guaranteeing convergence in the final phase where the optimization step size is already small and variance dominates.
>
> To further verify this effect, we additionally conducted experiments comparing the cases where r is gradually reduced versus kept fixed. On the SST-2 dataset with the OPT-2.7B model, the convergence curves of the two settings almost overlap in the early stage of training: both reach a loss of approximately 0.34 which is comparable to MeZO’s loss after 4,000 steps within only about 1100-1200 steps. However, in the mid-to-late training stage, the curves diverge. When r is kept fixed, the loss begins to rise, reaching 0.42 at 2,000 steps and further increasing to 0.53 at 4,000 steps. In contrast, when r is gradually reduced, the loss continues to decrease and reaches a significantly lower value of 0.27 at 4,000 steps. These results indicate that shrinking r helps maintain training stability in the later stages, while having little to no negative effect on early-stage convergence.
>
>
>
> **[Budget normalization and hyper parameters]**
>
> For clarity: MeZO uses 1 perturbation direction in all tables. HiZOO also uses 1 direction. KerZOO uses 3 perturbations, matching the configuration used in our derivation. The sensitivity for kernel parameters β and C is already provided in Table 5 and Table 6, showing that the method is robust over a wide range of kernel settings. Regarding n, we have done some experiments on OPT-2.7B, SST-2, summarizing the convergence loss for different n: n = 2: loss is about 0.31,  n = 3: loss is about 0.28 and n = 4: loss is about 0.26. This shows that n = 3 already achieves strong performance, and increasing n may further increase total training time. All baseline results in Tables 1–3 use one perturbation, which is the standard baseline configuration. Importantly, a single perturbation of baseline already takes longer wall-clock time than one KerZOO update, even though KerZOO uses multiple perturbations. Because our objective is efficiency and faster convergence without accuracy drop, the meaningful fairness criterion is total training time, not artificially equalized “forward-passes per update.” Under the same or less total training time, KerZOO consistently reaches equal or higher accuracy and lower loss than baseline, and this is exactly the efficiency gain our method is designed to provide.
> The βₜ schedule mentioned in Algorithm 1 is not a learning rate nor a shrink parameter.
> It is simply the momentum weight used in an accelerated SGD update, following the formulation in [1].
>
> [1] Accelerated Zero-Order SGD under High-Order Smoothness and Overparameterized Regime (arXiv:2411.13999).

---

### Official Review · Reviewer_PJ43 · 2025-11-02

**Soundness:** 3
**Presentation:** 3
**Contribution:** 3
**Rating:** 6
**Confidence:** 3

**Summary:**

KerZOO addresses the slow convergence problem in zeroth-order (ZO) optimization for LLM fine-tuning by introducing a kernel function to reduce gradient estimation bias. The paper analytically identifies that standard ZO methods suffer from lower-order bias due to random perturbations, which significantly hinders convergence speed. By designing kernel functions based on Legendre polynomials that satisfy specific moment conditions, KerZOO eliminates this second-order bias term and improves estimation accuracy to higher-order terms only. Experiments across RoBERTa, OPT, and LLaMA models demonstrate that KerZOO achieves up to 74% reduction in GPU training hours while improving accuracy by 2-3% compared to the MeZO baseline. The method is compatible with parameter-efficient fine-tuning techniques like LoRA and maintains the memory efficiency advantages of zeroth-order optimization.

**Strengths:**

1. The research problem in this paper is very important and interesting. ZO methods usually need more iterations to converge than first-order methods, so accelerating the training process will be important and interesting.

2. The proposed method in this paper is very easy to follow and the kernel-based method is very interesting.

3. The paper provides thorough experiments to verify the performance gain of the proposed method KerZOO.

**Weaknesses:**

1. I think the main concern is from the experiments. I noticed the paper provides the results on LLaMA3, but most experiments focus on RoBERTa and OPT. I hope the authors can provide more results on the state-of-the-art pre-trained models. Because I also think a strong pre-trained model can narrow the performance gap between zeroth-order based methods and first-order based methods.

2. I think the paper focused on accelerating the ZO training process, ans some related paper also focused on this direction, the authors can also provide these comparison results, such as SensZOQ, Sparse MeZO, ZO-SVRG.

[1] Zeroth-Order Fine-Tuning of LLMs with Transferable Static Sparsity.
[2] Sparse MeZO: Less Parameters for Better Performance in Zeroth-Order LLM Fine-Tuning.
[3] Zeroth-order stochastic variance reduction for nonconvex optimization.

**Questions:**

1. In my opinion, ZO method is very sensitive to the selection of hyper-parameters, like learning rate. I would like to ask whether the authors fully tune the hyper-parameters.

---

> ### Author Response · Authors · 2025-11-26
>
> **[More experiments about SOTA model]**
>
> We appreciate the reviewer’s suggestion.
> We would like to clarify that our paper already includes experiments on two state-of-the-art LLaMA3 models (LLaMA3-3B and LLaMA3-8B) in Figure 2. These results show that KerZOO achieves up to about 10% accuracy improvement over MeZO and HiZOO while requiring fewer iterations.
> Thank you for this valuable suggestion.
> In addition to the LLaMA3-3B and LLaMA3-8B results already shown in Figure 2 of the main submission, we have further conducted additional experiments on LLaMA3-3B, following the reviewer’s request for more results on strong pre-trained models.
> Specifically, on the MultiRC dataset: MeZO: 64.8 HiZOO: 64.3 KerZOO (ours): 65.2
> On the BoolQ dataset: MeZO: 78.0 HiZOO: 78.8 KerZOO (ours): 78.2
> In addition, we have included convergence–time comparisons on LLaMA-3-3B between KerZOO and the MeZO baseline.
> | Dataset | MeZO Convergence Time | KerZOO vs MeZO (%) |
> |--------|------------------------|--------------------|
> | SST-2  | 100% | **17.5% of MeZO** |
> | CB     | 100% | **16.7% of MeZO** |
> | WSC    | 100%  | **33.3% of MeZO** |
> | WiC    | 100% | **13.6% of MeZO** |
>
> We will incorporate these new LLaMA3 results into the final version.
>
>
> **[More works on accelerating the ZO training process]**
>
> We thank the reviewer for highlighting these additional zeroth-order optimization directions.
> We would like to clarify that Sparse MeZO is already included in our experimental comparison.
> Table 16 in the appendix reports detailed results, and KerZOO consistently outperforms Sparse MeZO across multiple datasets under comparable computational budgets.
> We will make this comparison more visible in the revised version.
>
> | Datasets   | SST2 |  RTE  | SQuAD | WiC  | BoolQ |
> |------------|------|-------|-------|------|-------|
> | Sparsezero | 92.3 | **76.9** | 77.9  | 58.2 | **76.5** |
> | Subzero    | 92.1 | 74.0  | 84.5  | 60.8 | 75.3  |
> | KerZOO     | **92.6** | 72.8  | **85.0**  | **64.2** | 76.2  |
>
> ZO-SVRG is a theoretical optimization result for zeroth-order methods and has not been applied to LLM fine-tuning. Instead, we adopt a separate baseline, MeZO-SVRG [1], which incorporates the SVRG idea into MeZO for LLM training. We compare against this baseline: for example, on RoBERTa-large, MeZO-SVRG achieves 84 accuracy on SST-2 and 49 on MNLI, while KerZOO achieves 92.1 and 63.8 respectively.We will add these results to the appendix in the final version of the paper.
>
> SensZOQ is a memory-efficient extension of ZO fine-tuning that applies sparsification and quantization so that ZO methods can run on on-device hardware. In contrast, KerZOO improves the ZO optimizer itself, reducing overall convergence time. Thus, the two methods are orthogonal: KerZOO can potentially be combined with SensZOQ, although a deeper investigation is needed to determine which optimization parameters, especially the sparse vs. dense components in SensZOQ, can benefit from KerZOO’s kernel-based estimator.
>
> [1] Gautam, Tanmay, et al. "Variance-reduced zeroth-order methods for fine-tuning language models." arXiv preprint arXiv:2404.08080 (2024).
>
> **[About the hyper parameters]**
>
> Thanks for reviewer's comment.
> We would like to clarify that we did not tune the learning rate or the perturbation radius ε for KerZOO, they are kept identical to the MeZO settings to ensure a strictly fair comparison.

---

### Note · Program_Chairs · 2026-01-17
**Submission Desk Rejected by Program Chairs**

The following references in this submission do not refer to real documents and/or have major errors in bibliographic information:

 Marie-Catherine De Marneffe, Nicolas Simard, Wanrong Xu, et al. The shared task on implicit and explicit hate speech detection. In W-NUT, 2019.
Xiangru Yao and Jimmy Lin. Improving retrieval-based sentence generation with context-aware answer selection.  In AAAI, 2020.